# Orientation Matters: Making 3D Generative Models Orientation-Aligned

Yichong Lu[1,2*]    Yuzhuo Tian[1*]    Zijin Jiang[1*]    Yikun Zhao[1]    Yuanbo Yang[1,2]

Hao Ouyang[2]    Haoji Hu[1]    Huimin Yu[1]    Yujun Shen[2]    Yiyi Liao[1†]

[1]Zhejiang University    [2]Ant Group

https://xdimlab.github.io/Orientation_Matters

## Abstract

Humans intuitively perceive object shape and orientation from a single image, guided by strong priors about canonical poses. However, existing 3D generative models often produce misaligned results due to inconsistent training data, limiting their usability in downstream tasks. To address this gap, we introduce the task of orientation-aligned 3D object generation: producing 3D objects from single images with consistent orientations across categories. To facilitate this, we construct Objaverse-OA, a dataset of 14,832 orientation-aligned 3D models spanning 1,008 categories. Leveraging Objaverse-OA, we fine-tune two representative 3D generative models based on multi-view diffusion and 3D variational autoencoder frameworks to produce aligned objects that generalize well to unseen objects across various categories. Experimental results demonstrate the superiority of our method over post-hoc alignment approaches. Furthermore, we showcase downstream applications enabled by our aligned object generation, including zero-shot object orientation estimation via analysis-by-synthesis and efficient arrow-based object rotation manipulation.

## 1  Introduction

Humans possess a remarkable ability to imagine the 3D structure of an object ("synthesis") and infer its properties, such as orientation ("analysis"), from a single image. We intuitively recognize the correct orientation of a car on the road, know how to grasp a cup by its handle, or decide how to place a chair in a room. In cognitive science, this ability is linked to the concept of *object constancy*—the capacity to mentally reconstruct a canonicalized object despite changes in viewpoint or other variations [26, 16]. This synthesis of an internal object representation enables analysis of its pose, a process often described as analysis by synthesis.

Replicating this perceptual capability has long been a central pursuit in computer vision. Recent progress in generative AI has significantly advanced single-view 3D object generation, making it both accessible and effective. This advancement in "synthesis" offers a rapid and adaptable means of constructing 3D models from everyday images, enabling a wide range of applications in AR/VR content creation, robotic simulation, etc.

Despite these strides, a critical challenge remains: *orientation alignment*. Existing methods often neglect this aspect, largely due to inconsistencies in large-scale 3D datasets [6, 57]. As a result, generated models frequently lack standardized canonical orientations—chairs may face different directions, mugs may appear tipped over, and vehicles may be misaligned. This disparity between human perceptual consistency and the output of generative models poses a significant obstacle for

---

*Equal contribution. †Corresponding author.

39th Conference on Neural Information Processing Systems (NeurIPS 2025).

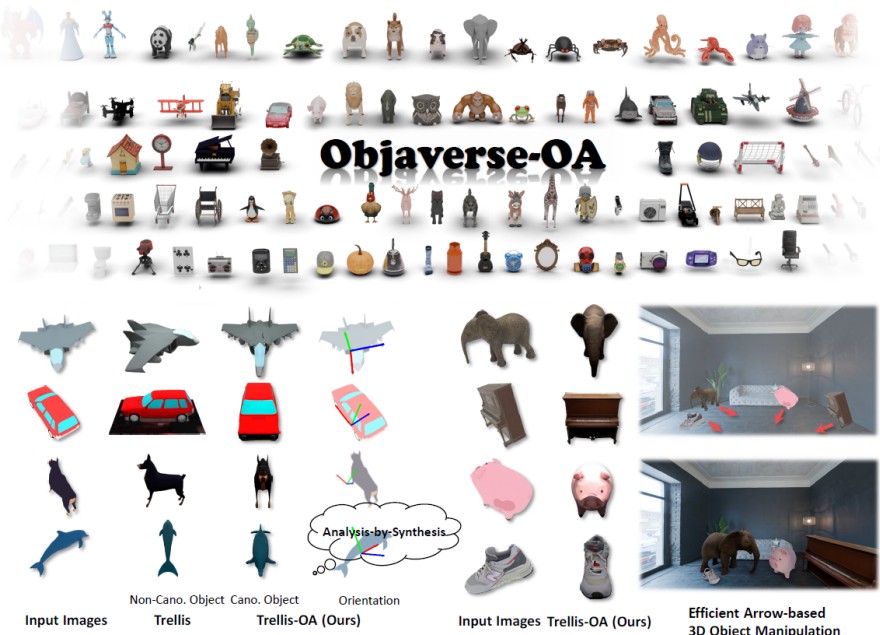

Figure 1: **Objaverse-OA for Orientation-Aligned Generation**. We construct a new dataset named Objaverse-OA, which contains orientation-aligned 3D models across 1008 categories (top). Using Objaverse-OA, we make existing 3D generative models orientation-aligned, which can further be used for zero-shot model-free orientation estimation (bottom left) and efficient arrow-based 3D object rotation manipulation (bottom right).

downstream tasks such as analysis-by-synthesis, e.g., orientation estimation. To bridge this perceptual gap, it is essential that 3D generative models not only reconstruct object geometry but also generate objects in consistent, semantically meaningful orientations.

In this paper, we introduce a novel task: orientation-aligned 3D object generation—the generation of a 3D object from a single image such that its orientation is consistently aligned both within and across categories, in accordance with common-sense priors. While no existing methods explicitly address this task, a possible workaround involves first generating a 3D object and then applying pose estimation to align its orientation. However, achieving robust and generalizable orientation alignment across diverse object categories remains a significant challenge. Most 3D pose estimation approaches [54, 29, 40] focus on predicting **relative** poses with respect to predefined 3D CAD models or multi-view reference images. In contrast, our task demands **absolute** orientation prediction — aligning objects to a canonical coordinate frame that corresponds to intuitive understandings of front, top, and upright directions. While some prior works explore absolute pose estimation [47, 3], they are typically constrained to a limited set of categories, primarily due to the substantial manual effort required to curate orientation-aligned training data. Recent efforts to scale such methods have relied either on labor-intensive human annotations [15, 24] or on the use of Vision-Language Models (VLMs) [52]. However, the former approach remains category-restricted, while the latter is subject to the inherent inaccuracies of VLM-based orientation predictions. Moreover, these strategies necessitate complex post-processing after 3D generation, which not only introduces potential inaccuracies but also reduces user-friendliness and practical applicability.

To address this, we propose to learn canonical 3D object generation by directly fine-tuning pre-trained 3D generative models, thereby avoiding the limitations of such two-stage pipelines. Our key insight is that a sufficiently diverse set of orientation-aligned 3D models can effectively adapt existing generative models from producing arbitrarily oriented outputs to generating objects with consistent, canonical orientations, while maintaining strong generalization capabilities to unseen objects across various categories. To support this, we introduce Objaverse-OA, a new dataset comprising 14,832 3D models spanning 1,008 categories, each aligned to a consistent, common-sense orientation. Leveraging Objaverse-OA, we fine-tune two representative 3D generative models [57, 23] to produce

Trellis-OA and Wonder3D-OA, allowing for generating well-aligned 3D objects across a broad spectrum of categories, including those not included in the fine-tuning set.

We further demonstrate the utility of orientation-aligned 3D generative models through two downstream applications: zero-shot 3D object orientation estimation and efficient arrow-based object rotation manipulation. For orientation estimation, our canonically generated 3D models serve as templates to estimate object poses from single images, generalizing well across categories. Moreover, we develop a user-friendly interface for object rotation manipulation in the augmented reality applications and 3D software, allowing users to specify the desired orientation via drawing an arrow, thereby facilitating precise placement without tedious pose adjustments.

Our contributions are as follows: 1) We introduce the novel task of orientation-aligned 3D object generation across a wide range of categories. 2) We construct Objaverse-OA, the largest orientation-aligned 3D dataset in terms of category coverage. 3) We fine-tune existing 3D generative models on Objaverse-OA to enable canonical object generation with robust generalization to unseen objects across various categories. Experimental results across multiple datasets demonstrate that our method achieves superior orientation alignment compared to existing baselines. 4) We showcase the practical benefits of our orientation-aligned models in two key applications: zero-shot orientation estimation and efficient object rotation manipulation via intuitive user interaction.

## 2 Related Work

**3D Generative Models:** Early 3D generation methods [55, 36, 1, 28, 4] typically employed GANs [46] to model 3D distribution, while these methods generate orientation-aligned objects, they are limited to a single category. The recent breakthroughs in 2D diffusion models [5, 11] provide new solutions for 3D generation. Pioneering works DreamFusion [33] and SJC [48] propose to generate 3D models by distilling from a 2D text-to-image generation model. However, these methods and their follow-ups [18, 19, 43, 42, 51] always suffer from low efficiency and multi-face problems due to per-shape optimization and lack of explicit 3D supervision. Recently, methods based on multi-view diffusion models [23, 21, 49, 37, 41, 38, 44, 60] have succeeded in efficiently producing multi-view consistent images via 3D attention. More recently, [25, 57, 64, 62, 59] utilize 3D latent spaces to further improve the geometry quality of the 2D-assisted approaches. However, despite huge progress on quality and efficiency, they all produce 3D models with uncanonical orientations due to orientation misalignment in their 3D training data, like Objaverse [6], Objaverse-XL [7], and TRELLIS-500K [57].

**Object Orientation Estimation:** One feasible approach to align 3D model orientations is to render the 3D models from a fixed camera and estimate the object orientations in the renderings. Although image-based 3D object pose estimation has been widely researched, most methods [54, 29, 40, 17, 22, 61] focus on predicting relative poses based on known 3D CAD models or reference images. Since the orientation-aligned 3D CAD models and reference images are not available, they cannot estimate the object orientations aligned with common sense. Category-level object pose estimation methods [47, 3, 53, 20, 9, 45] address this problem by generating 3D shape priors from the input images. However, most of them are limited to category-level due to heavy labor effort in constructing orientation-aligned 3D datasets [47, 2, 15, 24]. In contrast, our method can generate orientation-aligned 3D objects across a large number of categories. Concurrently, Orient Anything [52] realizes zero-shot object orientation estimation by automatically constructing large orientation-aligned 3D datasets using advanced Vision Language Models (VLMs). However, it still suffers from generalizability and accuracy due to a lack of training on the object orientation estimation task. Besides, post-processing after 3D generation is costly compared to directly generating orientation-aligned 3D models. Concurrently, [13] proposes an intra-category object pose canonicalization method and constructs Canonical Objaverse Dataset, which contains 3D objects with canonical poses within categories. However, our work focuses on inter-category object pose canonicalization.

## 3 Objaverse-OA Dataset

In this section, we introduce the construction of our dataset, Objaverse-OA. Dataset diversity plays a crucial role in achieving strong generalization capability. To the best of our knowledge, the existing orientation-aligned 3D dataset [24] with the largest category number includes only 200 categories

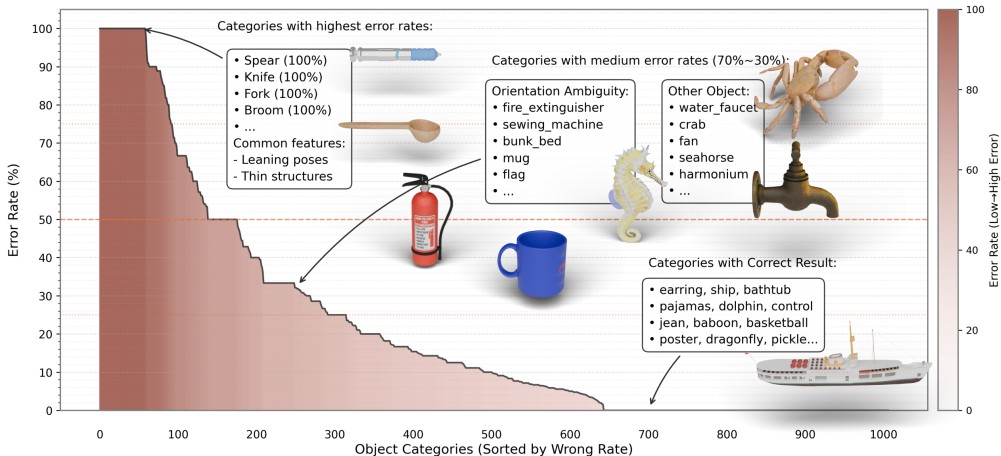

Figure 2: **VLM's Performance in Orientation Estimation**. We utilize our manually curated dataset as ground truth (GT) and show the error rate of VLM's estimation across different categories. We observe that (1) the VLM demonstrates particular difficulty in recognizing front-facing orientations for stick-like objects, and (2) a significant portion of recognition errors occur when processing objects with inherently unclear or ambiguous frontal views. These challenges highlight the necessity of our manual curation.

and fewer than 2,000 3D objects. In contrast, our Objaverse-OA dataset contains 14,832 orientation-aligned 3D objects across 1008 categories, which will be made publicly available to the research community. To build this large-scale dataset while maintaining both efficiency and accuracy, we employ a hybrid pipeline that combines Vision-Language Model (VLM) pre-processing with manual correction.

**VLM pre-processing:** As discovered by Orient Anything [52], advanced VLMs demonstrate the ability to recognize object front views without task-specific training. Since most models in Objaverse primarily vary in the horizontal (yaw) axis, we follow the strategy proposed in Orient Anything: we render each 3D object from four horizontal viewpoints—front, back, left, and right—and use a VLM to identify the correct front view. Based on the identified view, we then rotate the 3D model accordingly to align it to a canonical orientation. Our data processing begins with the Objaverse-LVIS dataset, and we use Gemini-2.0 [32] as the VLM for view recognition. From a total of 46,219 3D models, Gemini successfully identifies front views for 20,664 objects. However, we observe that VLM-based recognition, while promising, still falls short of human-level accuracy due to the absence of fine-tuning and challenges associated with ambiguous or difficult cases. We illustrate the error rate of VLM's prediction across different categories in Fig. 2. One of the challenging cases is stick-like objects, like spears, keys, and forks, since many of them are not aligned in roll and pitch angles in the Objaverse. Another challenge involves geometrically narrow or thin objects such as fish, bicycles, and water faucets. In such cases, VLM struggles to identify the correct orientation, as it relies solely on visible front-view features without reasoning from side-view context, unlike humans. What's more, objects with ambiguous front views, like teapots, extinguishers, and cups, can result in inconsistent orientation predictions. To address these issues, we introduce a manual correction step described below.

**Manual correction:** Starting from the VLM-based alignment, we manually filter or correct the wrong recognition results of the VLM and canonicalize objects with the ambiguous front-view definition. Moreover, to preserve category diversity, we reintroduce objects that were incorrectly filtered out by the VLM, particularly in cases where a category has too few correctly aligned instances. As illustrated in the Figure 2, we manually correct the orientations for about 600 object categories, especially the stick-like objects and objects with ambiguous orientation or unclear front view features. Note that some objects have orientation ambiguity, especially tools like spoons, mugs, and fire extinguishers. During the manual correction process, we refer to the object orientations defined in prior work, specifically ImageNet3D [24], for the categories it covers. For example, for spoons,

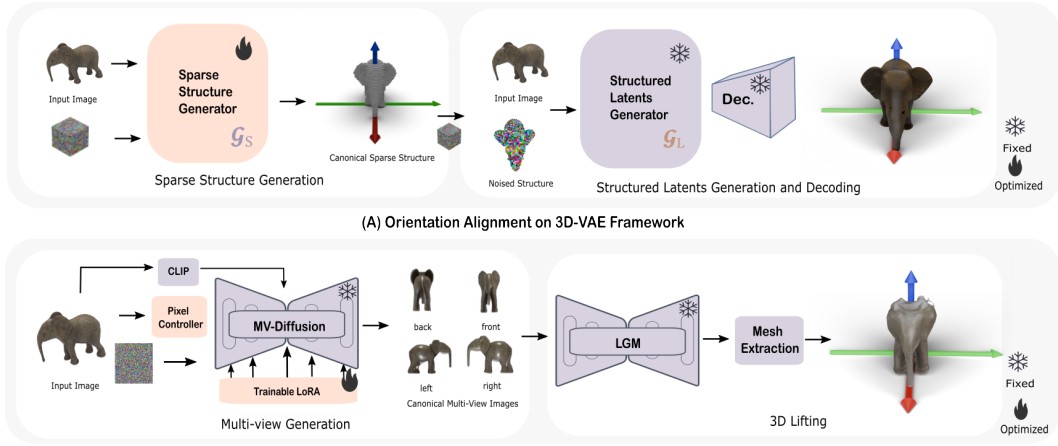

Figure 3: **Trellis-OA and Wonder3D-OA.**We fine-tune two representative methods: Trellis [57], based on a 3D-VAE backbone (top), and Wonder3D [23], based on a multi-view diffusion backbone (bottom). For the 3D-VAE, we find that fine-tuning only the sparse structure generator is sufficient to produce orientation-aligned objects. For the multi-view diffusion model, we adopt LoRA as a lightweight domain adapter to enable the generation of orientation-aligned target images.

which are included in ImageNet3D, we align their poses in our dataset accordingly. For ambiguous objects only included in our dataset, we define canonical poses based on semantic part structures, geometric features, and common knowledge, following the principles established by ImageNet3D and our supplementary material Section A.1. What's more, we also filter objects with low geometry quality and scenes with multiple objects. Experiments show that the pre-trained 3D generative models can further improve geometry quality after fine-tuning on our dataset.

## 4 Orientation-Aligned 3D Object Generation

Based on our curated Objaverse-OA (Section 3), we fine-tune existing 3D generative models to generate orientation-aligned objects. To demonstrate that orientation-aligned object generation benefits a variety of architectural backbones, we implement our approach on two widely used single-view image-to-3D reconstruction frameworks: a 3D VAE-based generative model (see Section 4.1) and a multi-view diffusion model (see Section 4.2).

### 4.1 3D-VAE Based Generative Model

We choose a state-of-the-art 3D-VAE-based generative model, Trellis [57], as our base model, which can produce fine-grained geometry and appearance aligned with the input image.

**Preliminary:** As shown in the Figure 3, Trellis [57] adopts three modules for 3D asset generation during inference including sparse structure generator $\mathcal{G}_S$, structured latents generator $\mathcal{G}_L$, and 3D decoder $\mathcal{D}$. Given the input image $\mathbf{I}$, sparse structure generator $\mathcal{G}_S$ produces dense binary 3D grid $\boldsymbol{O} \in \{0,1\}^{N \times N \times N}$: $\boldsymbol{O} = \mathcal{G}_S(\mathbf{I}, \boldsymbol{\varepsilon}_{3d})$, where N is the length of the grid and $\boldsymbol{\varepsilon}_{3d}$ is the 3D noise sampled from $\mathcal{N}(0,1)$, which is further converted into active voxels $\{(\boldsymbol{p}_i)\}_{i=1}^N$ defined as sparse structure. After that, sparse latents generator $\mathcal{G}_S$ is used to generate structured latents $\boldsymbol{z} = \{(\boldsymbol{z}_i, \boldsymbol{p}_i)\}_{i=1}^N$: $\boldsymbol{z} = \mathcal{G}_L(\mathbf{I}, \boldsymbol{z}_{noised})$, where $\boldsymbol{z}_{noised}$ is the noised sparse structure $\{(\boldsymbol{\varepsilon}_i, \boldsymbol{p}_i)\}_{i=1}^N$. Finally, the 3D representation $\mathcal{M}$ is obtained via 3D decoding: $\mathcal{M} = \mathcal{D}(\boldsymbol{z})$.

**Trellis-OA:** The original Trellis model [57] is unable to produce orientation-aligned 3D outputs due to the orientation inconsistencies present in its training data. To address this, we fine-tune Trellis using our Objaverse-OA dataset, resulting in Trellis-OA, which generates 3D objects with aligned orientations. Although Trellis comprises several modules for 3D generation, we find that fine-tuning only the sparse structure generator $\mathcal{G}$S is sufficient for achieving orientation alignment. This is likely because Trellis inherently generates object poses randomly sampled from four orthogonal directions, and our aligned pose distribution resides within this range. As a result, both the pre-trained structured latent generator $\mathcal{G}$L and the 3D decoder $\mathcal{D}$ remain compatible with the aligned orientations and do

not require additional fine-tuning. Specifically, our fine-tuned sparse structure generator $\mathcal{G}'_{\mathrm{S}}$ generates canonical sparse structure $\{(\boldsymbol{p}'_i)\}_{i=1}^{N}$, which shares aligned orientations. Afterwards, pre-trained structured latents generator $\mathcal{G}_{\mathrm{L}}$ generates canonical structured latents and 3D decoder $\mathcal{D}$ produces final canonical 3D models $\mathcal{M}_{OA}$ with orientation aligned. Our experiments demonstrate that we can efficiently fine-tune Trellis to an orientation-aligned one while preserving its 3D priors.

## 4.2 Multi-view Diffusion Model

We choose Wonder3D [23] as our base multi-view diffusion model since it is one of the most representative works based on multi-view diffusion frameworks. Note that the recipe for Wonder3D can also be used by other multi-view diffusion based methods.

**Preliminary:** Multi-view diffusion models are typically fine-tuned from large-scale pre-trained text-to-image models [35] and achieve multi-view consistent via 3D-aware attention mechanisms. Specifically, given an input image $\mathbf{I}$, the multi-view diffusion model $\boldsymbol{MV}$ generates $N$ multi-view images $\{\mathbf{I}_{mv}^{i}\}_{i=1}^{M} = \boldsymbol{MV}(\mathbf{I}, \varepsilon)$, where $\varepsilon \in \mathcal{N}(0,1)$ is a sampled noise vector. Here, the poses of the ground truth of the multi-view images $\{\mathbf{I}_{mv}^{i}\}_{i=1}^{N}$ are dependent on the input image $\mathbf{I}$, where $\mathbf{I}_{mv}^{1}$ is set to predict the input image $\mathbf{I}$. After that, Wonder3D adopts an optimization method based on NeuS [50] to lift $\{\mathbf{I}_{mv}\}$ to 3D representation $\mathcal{M}$.

**Wonder3D-OA:** Wonder3D [23] cannot produce orientation-aligned 3D models due to its misaligned training data and input-image related camera settings. To address this, we implement Wonder3D-OA, leveraging our orientation-aligned Objaverse-OA dataset. Given an input image $\mathbf{I}$, Wonder3D-OA first generates orientation-aligned, multi-view consistent images $\mathbf{I}_{OA}^{mv}$, which are then lifted to orientation-aligned 3D representations $\mathcal{M}_{OA}$. Our key change is in the supervision setup: instead of using input-view-dependent ground truth, we render six canonical views (front, front-left, front-right, left, right, back) from fixed camera poses based on Objaverse-OA. This provides consistent orientation references. We adopt LoRA [12] as a lightweight domain adapter to fine-tune the pre-trained Wonder3D, preserving its learned 3D priors from non-canonical data while enabling alignment. Additionally, we make architectural adjustments for improved performance. Originally, Wonder3D aligns the input image with one of the predicted views to inject local features, which fails to work in our canonical camera setting. Inspired by ImageDream [49], we instead employ a pixel injector to integrate local features into the multi-view diffusion model. Specifically, Wonder3D employs a 3D dense self-attention mechanism with a shape of $(b_z, 6, c, h_l, w_l)$ across six views within a transformer layer, where $b_z$ is the batch size, c is the number of feature channel, $h_l$ and $w_l$ are the image resolution. Our pixel controller modifies this to $(b_z, 7, c, h_l, w_l)$, incorporating the input image as an additional view. We further improve efficiency by replacing Wonder3D's test-time optimization-based 3D lifting module (NeuS) with LGM [44], a recent model tailored for sparse-view 3D reconstruction. Since LGM is trained using 4-views as input, we feed the front, left, right, and back views into LGM for 3D lifting. Finally, as our lifting module performs effectively without the need for normal maps, we omit fine-tuning the cross-domain attention module in Wonder3D, simplifying the pipeline without sacrificing quality.

## 5 Downstream Applications

To further demonstrate why making the 3D generative models orientation aligned is important, we implement two downstream applications, including zero-shot model-free object orientation estimation Section 5.1 and efficient arrow-based object rotation manipulation Section 5.2.

### 5.1 Zero-shot Model-free Object Orientation Estimation

One popular class of solutions to 3D object orientation estimation is based on analysis-by-synthesis [47, 3, 10], but most of them are limited to category-level due to the difficulty in synthesizing orientation-aligned 3D models across categories. Another line of methods [29, 54] estimates poses relative to a given CAD model, limiting practical use since a corresponding model must be available for each input image. While one could replace CAD models with generated 3D shapes (e.g., from Trellis), the misaligned outputs of existing generative models result in unreliable pose references. In contrast, our method generates orientation-aligned 3D shapes directly from a single image, enabling the prediction of *absolute poses* in a canonical frame by treating the generated 3D shape as

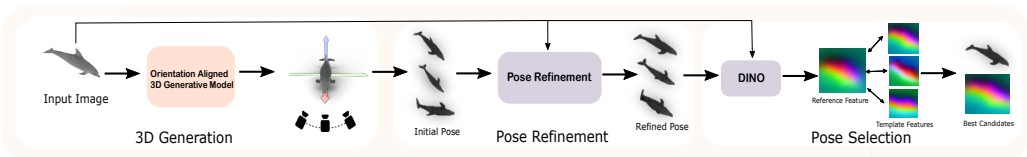

Figure 4: **Zero-Shot Orientation Estimation**. Our orientation-aligned 3D object acts as a template for pose estimation by rendering it from multiple views, refining each, and selecting the best-matching viewpoint. Note that we do not perform training for this downstream task, where the pose refinement module is directly from FoundationPose [54], and the pose selection module directly utilizes the pre-trained DINO feature extractor [27].

a per-object template. Specifically, we build on a state-of-the-art template-based pose estimation method, FoundationPose [54], which has strong generalizability due to training on a large synthetic dataset. It renders templates from a fixed set of viewpoints, refines each pose, and then selects the best match. Although FoundationPose is trained with accurate CAD models and depth maps, we find that its pose refinement module remains effective even with our generated 3D shapes and without depth input. However, its pose selection performance degrades when templates differ slightly in geometry or appearance. To address this, we retain the pose refinement module but replace the pose selection stage with a DINOv2 [27]-feature-based similarity metric, as shown in Figure 4. This is achieved by computing the L2 distance between DINOv2 patch feature maps of each refined rendering and the target image, and selecting the view with the highest similarity. Note that when estimating orientations of objects in scene-level images, we need to extract objects' masks via the image segmentation method [14] before 3D generation.

## 5.2 Efficient Arrow-based Object Rotation Manipulation

Efficient manipulation of 3D model rotation within simulation systems is crucial, yet challenging, especially when models are initialized in non-canonical poses. This difficulty arises because users typically aim to orient a model toward a desired direction, but conventional 3D simulation systems only record the model's pose relative to its initial state. As a result, users must manually compensate for any misalignment in the initial pose, complicating the interaction process. In contrast, our generated orientation-aligned 3D models enable a more direct and user-friendly rotation manipulation approach due to their aligned initial orientations. With this alignment, users can simply draw an arrow indicating the desired forward-facing direction, without needing to consider the model's original pose. This arrow-based interaction paradigm enhances usability and is compatible with both augmented reality (AR) applications and general-purpose 3D software. In AR applications, users draw an arrow in the 2D image plane, which is subsequently lifted into 3D space using monocular depth estimation techniques (e.g., [31]). The system then rotates the model so that its forward-facing axis aligns with the specified arrow, while ensuring the object remains grounded on the background plane. Note that because our models are normalized in scale, their size must still be specified, either through a large language model (LLM) or direct user input. In general 3D software, users draw the arrow directly in 3D space. The system then applies Rodrigues' rotation formula to align the model's orientation with the user-specified direction. Additional implementation details are provided in the supplementary materials Section A.3.

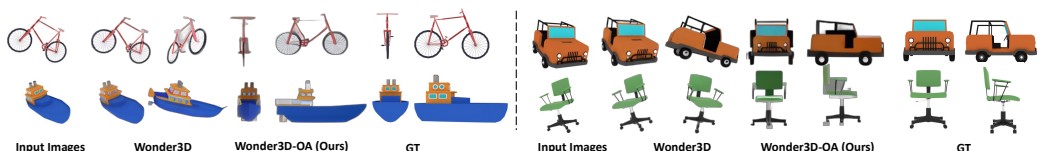

Figure 5: **Qualitative Results** on multi-view diffusion backbone, Wonder3D. For each input image, we show two views with the same index from the multi-view predictions.

| | GSO [8] | | | Toys4k [39] | | |
|---|---|---|---|---|---|---|
| | CD↓ | LPIPS↓ | CLIP↑ | CD↓ | LPIPS↓ | CLIP↑ |
| Wonder3D | 0.0894 | 0.2799 | 76.37 | 0.0932 | 0.2859 | 87.10 |
| Wonder3D + PCA | 0.0788 | 0.2554 | 77.80 | 0.0858 | 0.2691 | 87.58 |
| Wonder3D + VLM (Gemini-2.0) [32] | 0.0850 | 0.2752 | 76.30 | 0.0880 | 0.2804 | 87.53 |
| Wonder3D + Orient Anything (ViT-L) [52] | 0.1015 | 0.2600 | 77.50 | 0.1079 | 0.2699 | 88.12 |
| Wonder3D-OA (ours w/o LGM) | 0.0609 | 0.2300 | 80.22 | 0.0571 | 0.2351 | 91.33 |
| **Wonder3D-OA (ours)** | **0.0564** | **0.2270** | **80.30** | **0.0548** | **0.2317** | **92.09** |

Table 1: **Quantitative Comparison** of geometry and appearance on Multi-view Diffusion backbone [23]. We highlight the best, second-best, and third-best scores achieved on any metrics.

## 6    Experiment

### 6.1    Implementation Details

**Dataset.** Orientation-aligned 3D generative models Trellis-OA and Wonder3D-OA are trained on our Objaverse-OA dataset, which is curated from Objaverse-LVIS [6]. The base multi-view diffusion model is trained on Objaverse [6], and the base 3D-VAE-based model is trained on TRELLIS-500K [57]. To demonstrate the generalizability and accuracy of our method's orientation alignment ability, we evaluate on two unseen datasets, GSO [8] and Toys4k [39]. To further demonstrate the sim-to-real generalizability, we also evaluate on the real-world dataset Imagenet3D [24].

**Baselines.** For the task of aligned object generation, there are no existing baselines for this task. Therefore, we design baselines that perform this task in two stages: 1) object generation with misaligned orientations, and 2) orient them to aligned poses based on pose estimation using different variants: (i) Principal Component Analysis (PCA); (ii) advanced Vision Language Model (VLM) Gemini-2.0 [32]; and (iii) zero-shot model-free orientation estimation method, Orient Anything [52]. For the task of zero-shot orientation estimation, we compare our method with Orient Anything [52] and FSDetView [58]. Note that FSDetView doesn't support zero-shot estimation. Therefore, we evaluate its performance only on its supported categories.

**Metrics.** To evaluate the orientation alignment ability, we rotate reconstructed 3D models using different kinds of methods and calculate Chamfer Distance (CD), LPIPS [63], and CLIP [34] scores to measure the orientation alignment quality. To evaluate the performance of our zero-shot orientation estimation method, we calculate Acc@30 and orientation absolute error (Abs) according to the rotation error. We follow NOCS to calculate the rotation $e_R$ defined by: $e_R = arccos\frac{Tr(\tilde{R} \cdot R^T)-1}{2}$, where $Tr$ represents the trace of the matrix. Note that for stick-like objects, top and side directions typically have ambiguity. Therefore, we only calculate the rotation error in the front direction.

**Training and inference time.** To fine-tune Trellis-OA, we use a total batch size of 64 for training 30000 steps, which takes only about 10 hours on the cluster of 8 Nvidia Tesla A100 GPUs. To fine-tune Wonder3D-OA, we use a total batch size of 512 for training 40000 steps, which takes about 3 days on the cluster of 8 Nvidia Tesla A100 GPUs.. For the 3D object generation, Trellis-OA takes 32.93s and Wonder3D-OA takes 6.69s for multi-view generation, 2.53s for fused 3DGS generation, and 45.37s for mesh extraction. Given the generated 3D models, our pose refinement module takes 6.54s and the pose selection module takes 7.47s, which are comparable with the SOTA template-based pose estimation method FoundationPose [54].

### 6.2    Orientation-Aligned Object Generation

**Comparison with baselines:**   We first evaluate the performance of orientation-aligned object generation. As shown in the Table 1 and Table 2, our fine-tuned models, Wonder3D-OA and Trellis-OA, surpass most two-stage baselines in geometry and appearance on both GSO and Toys4k datasets. Note that our method is suboptimal in appearance on the GSO dataset compared to the VLM baseline. It is probably because the GSO dataset has more objects with irregular geometry and appearance, which is challenging for our fine-tuned generator to handle. We present more qualitative results in the Figure 5 and Figure 6. As shown in Figure 6, PCA cannot handle objects with different shape features and has difficulty in distinguishing the direction of estimated principal axes. VLM

| | GSO [8] | | | Toys4k [39] | | |
|---|---|---|---|---|---|---|
| | CD↓ | LPIPS↓ | CLIP↑ | CD↓ | LPIPS↓ | CLIP↑ |
| Trellis | 0.0770 | 0.2502 | 82.88 | 0.0951 | 0.2722 | 92.31 |
| Trellis + PCA | 0.0798 | 0.2771 | 78.84 | 0.0911 | 0.2758 | 88.09 |
| Trellis + VLM (Gemini-2.0) [32] | 0.0421 | **0.1998** | **89.97** | 0.0564 | 0.2137 | 95.19 |
| Trellis + Orient Anything (ViT-L) [52] | 0.0604 | 0.2193 | 88.61 | 0.0725 | 0.2313 | 94.81 |
| Trellis-OA (ours small) | 0.0448 | 0.2224 | 82.46 | 0.0465 | 0.2055 | 93.74 |
| Trellis-OA (ours) | **0.0407** | 0.2118 | 88.41 | **0.0393** | **0.1932** | **95.71** |

Table 2: **Quantitative Comparison** of geometry and appearance on 3D-VAE backbone [57].

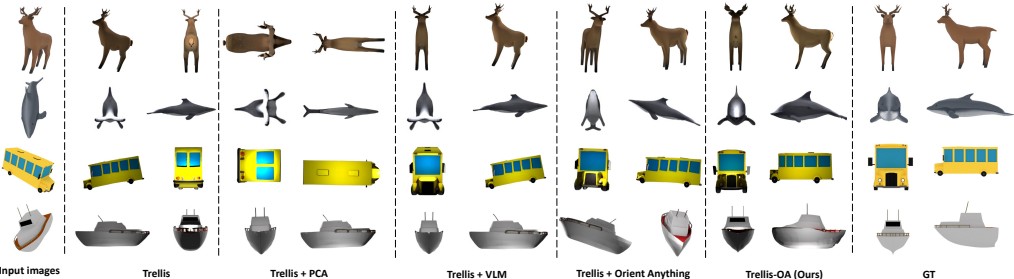

Figure 6: **Qualitative Results** based on the 3D-VAE backbone, Trellis. For each input image, we render the reconstructed object from two consistent views. Note that our method consistently generates objects in their canonical space.

has trouble recognizing objects with unclear front-view features, like dolphins and boats. Orient Anything is still not that accurate, and typically rotates objects into leaning poses. In contrast, our method accurately aligns the orientation of the generated 3D models without obvious geometry and appearance degradation. Besides, our method can handle real world internet images and multi-objects images as shown in Figure 7 and Figure 8. Furthermore, due to the high geometry quality of our Objaverse-OA, Trellis-OA eliminates the production of plane-like geometries. Please refer to the supplementary materials Section B for more qualitative results.

**Ablation Study:** To assess the importance of using an orientation-aligned dataset with diverse categories, we conduct an ablation study by fine-tuning 3D generative models on a reduced set of 100 categories with 5720 objects, see (ours small) in Table 1 and Table 2. The results show that reducing the number of training categories degrades both geometric and visual quality, highlighting the value of large category diversity in our Objaverse-OA dataset.

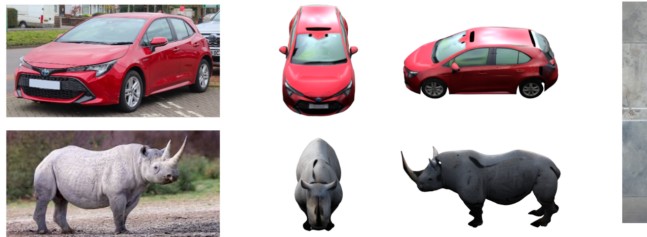

Figure 7: **Qualitative Results** of Trellis-OA on real world internet images.

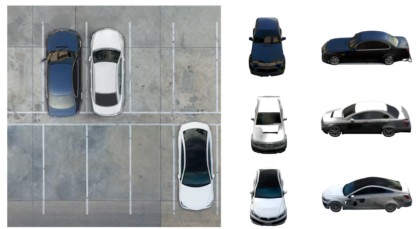

Figure 8: **Qualitative Results** of Trellis-OA on multi-objects images.

## 6.3 Zero-Shot Object Orientation Estimation

**Comparison with baselines:** As shown in Table 3, without specific training, our method is already comparable with the SOTA orientation estimation method [52] with the ViT-large architecture and surpasses [52] with the ViT-small architecture by a large margin. Besides, our method can handle challenging stick-like objects collected from the real-world dataset while baselines all fail to work, which further demonstrates our utility in coping with long-tailed situations.

|  | Toys4k [39] | | Stick-like Obj. from ImageNet3D [24] | |
| --- | --- | --- | --- | --- |
|  | Acc@30↑ | Abs↓ | Acc@30↑ | Abs↓ |
| FSDetView [58] (Few-shot) | 20.90 | 91.66 | 10.29 | 84.25 |
| Orient Anything [52] (Vit-S) | 42.05 | 52.72 | 2.63 | 81.70 |
| Orient Anything [52] (Vit-L) | **63.18** | **36.37** | 9.8 | 78.19 |
| Ours (Vit-S) | 51.15 | 53.94 | 60.78 | 40.12 |
| Ours (ViT-L) | 52.87 | 46.76 | **62.25** | **34.20** |

Table 3: **Quantitative Comparison** of zero-shot orientation estimation on Toys4k [39] and stick-like objects from ImageNet3D [24].

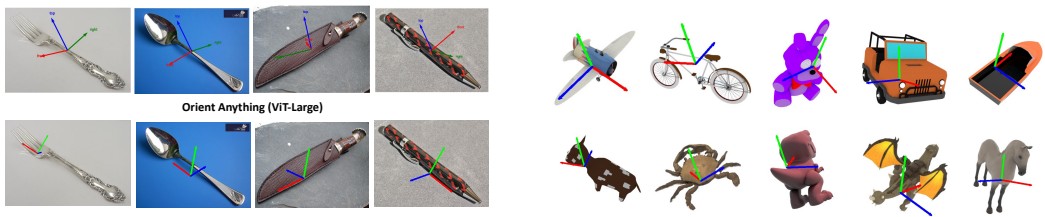

Figure 9: **Orientation Estimation Comparison** on stick-like objects from ImageNet3D [24].

Figure 10: **Orientation Estimation Results** of our method on Toys4k [39].

### 6.4 Efficient Arrow-based Object Rotation Manipulation

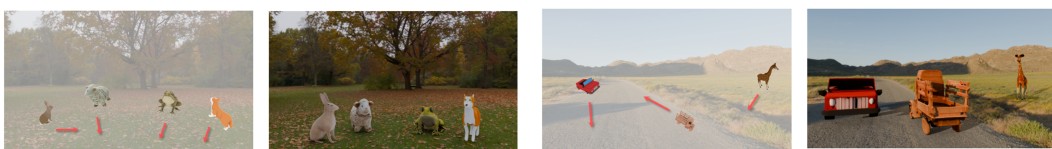

Figure 11: **Qualitative results** of our efficient arrow-based object rotation manipulation method in augmented reality applications.

Figure 1 and Figure 11 illustrate our arrow-based object rotation manipulation method in augmented reality applications. Given an image of an object, we generate the corresponding object in its canonical pose using Trellis-OA, and then manipulate it in the reference image with the target orientation indicated by a user-specified arrow. This allows for efficient object manipulation without adjusting the object's orientation in a post-hoc fashion. We present more qualitative results of our method in the general 3D software in the video of supplementary materials, which demonstrate the usage of our approach.

## 7 Conclusion and Limitation

In this paper, we aim to align the orientations of the 3D generative models for downstream orientation estimation and efficient object rotation manipulation in 3D simulation systems. Towards this goal, we construct Objaverse-OA, a dataset covering orientation-aligned 3D models across the largest number of categories. Based on Objaverse-OA, we align the orientations of 3D generative models based on two popular 3D generation frameworks, including multi-view diffusion and 3D-VAE. Built upon the orientation-aligned 3D generative models, we develop a simple but effective orientation estimation approach following the analysis-by-synthesis paradigm and an efficient arrow-based object manipulation method. While our method achieves promising zero-shot orientation estimation without task-specific training, future work could further improve performance by training a dedicated model that leverages the generated 3D objects as templates.

## 8 Acknowledgement

This work is supported by NSFC under grant 62441223 and 62202418, and partially by Ant Group.

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

# Appendix

In this appendix, we detail our dataset curation, experiment implementations, and method implementations in Section A. Besides, we provide more results of our efficient arrow-based object manipulation, orientation-aligned object generation, orientation estimation, and failure cases in Section B.

## A  Implementation Details

### A.1  Dataset Curation

**Data Processing Pipeline.**  We first apply Vision Language Model (VLM) recognition across the entire dataset, followed by manual identification and correction of failure cases. Specifically, four images were rendered for each 3D object using an orthogonal camera setup. The front view was identified using the VLM (Gemini 2.0 [32]), and the objects were subsequently rotated to their canonical poses based on the recognition results. Note that objects recognized as without front-view orientation are automatically excluded from the dataset. After this initial step, the orientations of the processed objects were manually refined using Blender software.

**VLM Pre-processing.**  Different from Orient Anything, we further enhance the text prompt by adding more recognition rules for different kinds of objects, as shown in Figure 13. We find this operation can further improve the robustness of VLM via in-context learning.

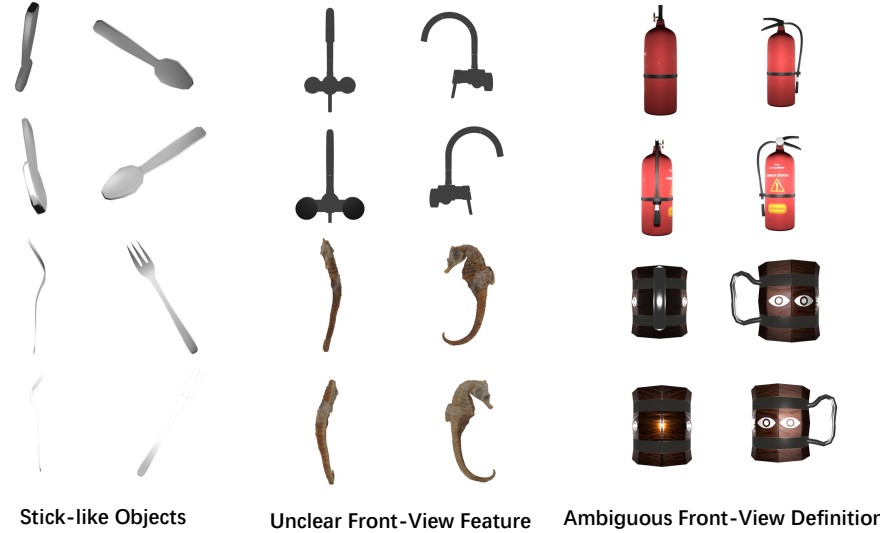

Stick-like Objects   Unclear Front-View Feature   Ambiguous Front-View Definition

Figure 12: Failure cases in VLM-based front-view recognition. The four images are rendered from four orthogonal cameras following our VLM-based recognition method.

**Manual Corrrection Details.**  Despite advancements, VLMs still face challenges in accurately recognizing objects' orientations due to a lack of training. Three particular object categories are prone to errors, as shown in Figure 12: (1) stick-like objects (e.g., forks, spoons), (2) objects lacking distinctive front-view features (e.g., seahorses, water faucets), and (3) objects with inherently ambiguous front-view definitions (e.g., mugs, fire extinguishers). VLMs struggle with stick-like objects since they are not aligned in roll and pitch angles. For objects without clear front-view features, distinguishing the front from the back often requires side-view inference, which hasn't been achieved by existing VLMs. Ambiguity in defining a canonical front view further complicates recognition for certain object types. To address these issues, we implemented the following corrections: stick-like objects were aligned parallel to the global X-axis, positioning the handle toward the negative X-axis and the functional end toward the positive X-axis. Objects without distinct front-view cues were manually rotated in Blender. For objects with ambiguous front-view definitions, we aligned the component typically associated with human interaction (e.g., handles) to the negative X-axis.

I'm going to show four images of the same object from four viewpoints in turn and label them 'A.' 'B.' 'C.' 'D.' Four options. The option 'E.' is " No front face or More than One front Face" . Decide whether it have a front and if yes, which one is the front of the object after the presentation. Note that: If the object is a gun, bow and arrow, etc., please use the muzzle of the gun as the front. Please note the following points: Bicycle: Front refers to the direction of the front, back refers to the direction of the rear, and side refers to both sides of the body. \Sofa: Front refers to the side with a backrest, back refers to the side without a backrest, and side refers to both ends of the sofa. \Pistol: Front refers to the direction of the muzzle, back refers to the direction of the grip, and side refers to both sides of the gun body. \Animals: with the front pointing towards the head, the back pointing towards the tail, and the side pointing towards both sides of the body. Person: Front refers to the face facing, back refers to the back facing, and side refers to the sides of the body. Car: Front refers to the direction of the front of the car, back refers to the direction of the rear of the car, and side refers to both sides of the car bodies. Watch: The front refers to the face of the dial, the back refers to the back of the watch, and the sides refer to both sides of the case. Teapot: The front refers to the direction of the spout, the back refers to the direction of the handle, and the side refers to both sides of the pot body; Cups: the front refers to the view when the handle is heading left; Other items: Please define the front, back, and side according to the structure and characteristics of the item. Key features: For each object, please consider its unique structure and features, such as the wheels of a bicycle, the backrest of a sofa, the muzzle of a handgun, the head of an animal, the face of a person, the headlights of a car, the dial of a watch, the spout of a teapot, etc. These features are usually the key to determining the orientation of an object. The reasoning process: you can simulate human thinking processes when determining the orientation of an object, such as first finding the most obvious feature on the object, then determining the orientation of that feature, and finally determining the overall orientation of the object. For some objects that are difficult to determine their orientation, multiple features can be combined for comprehensive judgment. Stick tools and weapons such as swords, axes, knives, and wrenches are considered to have no front. If you cannot decide or there is more than one front, you should choose 'E.' . If there are multiple possible choice, choose the one that has a higher confidence level of showing front view.

Figure 13: Text prompt for the Vision Language Model.

**VLM Error Analysis** After obtaining the orientation-aligned dataset, we conducted a comprehensive error analysis by comparing the VLM's predictions with our manually validated ground truth. For each 3D model, we computed the Chamfer Distance (CD) between the 3D models after VLM-based recognition and our manually corrected 3D models. Each model was uniformly sampled with 10,000 surface points to facilitate CD computation. Since CD values are influenced by object geometry and do not perfectly reflect orientation accuracy, we adopted a threshold-based approach to quantify recognition accuracy. Through empirical analysis, we found that rotating a model by 90° around its principal axis typically resulted in a CD > 0.01. This threshold, $\gamma = 0.01$, was therefore used to flag recognition errors: any model with a CD exceeding this value between VLM and ground-truth orientations was considered wrongly recognized. Notably, this threshold may produce wrong results for perfectly cylindrical or cubical shapes due to their inherent symmetry. However, such cases represent a negligible minority in our dataset and were therefore excluded from our error statistics to maintain analytical integrity.

## A.2 Experiment Implementations

**Evaluation Data.** We evaluate our method on three unseen datasets: GSO [8], Toys4k [39], and Imagenet3D [24]. For the GSO dataset, we randomly collected 48 objects with clearly defined front views. Each object was rendered into four images, sampled from the upper hemisphere with azimuth angles uniformly distributed within [0°, 360°], polar angles within [0°, 60°], and rotation angles within [-30°, 30°]. For the Toys4k dataset, we randomly selected 439 objects across 47 categories. Each object was rendered into a single image using the same camera configuration as applied to

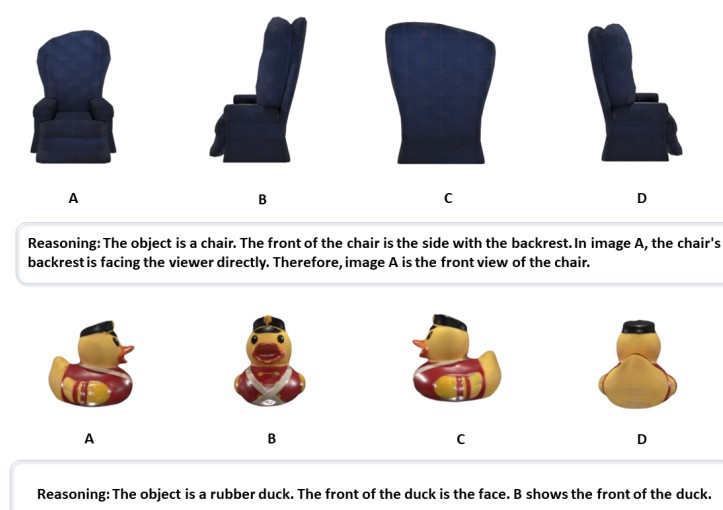

Reasoning: The object is a chair. The front of the chair is the side with the backrest. In image A, the chair's backrest is facing the viewer directly. Therefore, image A is the front view of the chair.

Reasoning: The object is a rubber duck. The front of the duck is the face. B shows the front of the duck.

Reasoning: The object appears to be a computer mouse. In image C, we can see the most prominent features of a mouse: the scroll wheel and buttons. Image A is most likely the back of the mouse. Images B and D show the side view. Thus, C is the front of the mouse.

Figure 14: Correct VLM recognition results.

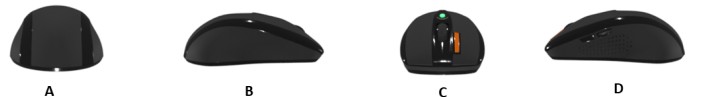

Reasoning: The object appears to be a spoon. The front of a spoon is the side of the bowl, so view B is the best candidate for the front view.

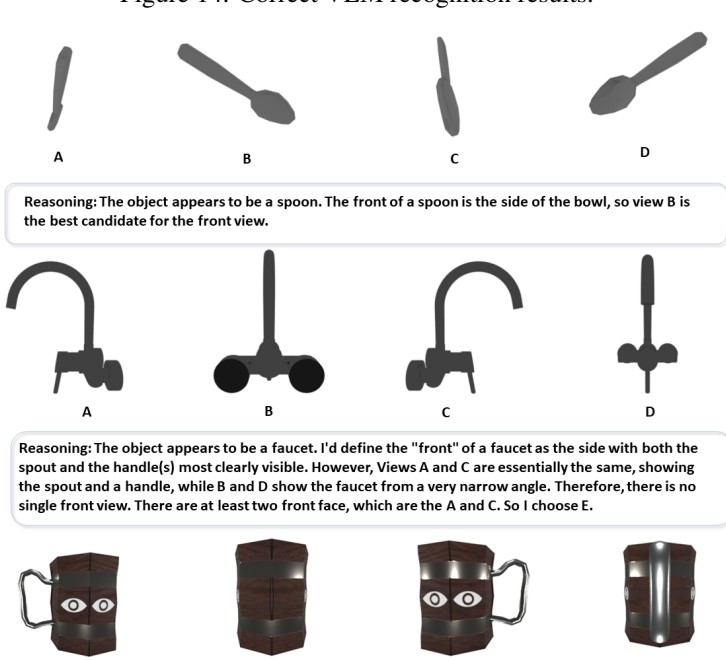

Reasoning: The object appears to be a faucet. I'd define the "front" of a faucet as the side with both the spout and the handle(s) most clearly visible. However, Views A and C are essentially the same, showing the spout and a handle, while B and D show the faucet from a very narrow angle. Therefore, there is no single front view. There are at least two front face, which are the A and C. So I choose E.

Reasoning: The object is a cup. Based on the definition, the front of a cup refers to the direction in which the handle of the cup faces backwards. A and C both showcase the handle, but the 'eye' design is better observed on A than C, making A a better choice as the front.

Figure 15: Wrong VLM recognition results.

the GSO dataset. For the Imagenet3D dataset, we focused on seven stick-like object categories, comprising a total of 204 objects. Please refer to Table 4 and Table 5 for details.

| airplane | bicycle | boat | bunny | bus | car | cat | chair |
|---|---|---|---|---|---|---|---|
| 9 | 10 | 9 | 9 | 5 | 5 | 9 | 20 |
| chicken | cow | crab | deer moose | dinosaur | dog | dolphin | dragon |
| 8 | 6 | 10 | 10 | 10 | 9 | 10 | 10 |
| elephant | fish | fox | frog | giraffe | guitar | helmet | helicopter |
| 10 | 10 | 9 | 10 | 10 | 10 | 7 | 8 |
| horse | laptop | lion | lizard | monkey | motorcycle | mouse | panda |
| 10 | 6 | 10 | 10 | 10 | 8 | 10 | 7 |
| PC mouse | penguin | piano | pig | radio | robot | shark | sheep |
| 3 | 16 | 6 | 10 | 2 | 8 | 10 | 8 |
| shoe | sofa | tractor | train | truck | violin | whale | |
| 10 | 20 | 16 | 4 | 10 | 10 | 9 | |

Table 4: Object numbers of each category in Toys4k [39] evaluation data.

| fork | knife | pen | rifle | scissors | screwdriver | spoon |
|---|---|---|---|---|---|---|
| 22 | 24 | 30 | 31 | 33 | 29 | 38 |

Table 5: Object numbers of each category in Imagenet3D [24] evaluation data.

**Baselines.** To implement the PCA-based baseline, we first calculate the three principal axes by eigen-decomposing the 3D models' geometries. The object is then rotated to align these principal axes with the global coordinate system's x-, y-, and z-axes. For the VLM-based baseline, we follow the same method used in the dataset curation, utilizing Gemini-2.0 [32] for orientation recognition. To implement the baseline based on Orient Anything [52], we use the official checkpoint based on the ViT-large architecture and adopt its data augmentation module. The object orientation is estimated from an image rendered using a fixed camera, and the object is subsequently rotated according to the predicted orientation. For the pre-trained 3D generative models based on multi-view diffusion and 3D-VAE, we utilize the official Wonder3D++ and Trellis checkpoints. For the orientation estimation baseline based on FSDetView [58], we restrict evaluation to the object categories supported by the method.

**Metrics.** For the calculation of Chamfer Distance, we don't perform PCA for both predicted 3D models and GT models (except the PCA baseline), in order to measure the pose canonicalization performance. As for the calculation of CLIP and LPIPS, for the 3D-VAE backbone, we render four orthogonal views with the camera elevation angle of 0 for both the generated and GT 3D models, and computed the LPIPS and CLIP scores based on these renderings at matched camera poses, while for the Multi-view Diffusion backbone, we randomly sample views on a unit sphere for evaluation to avoid unfairness, since the orthogonal views are aligned with the camera setting of our Wonder3D-OA, but not aligned with the Wonder3D baselines.

### A.3 Method Implementations

**Efficient Arrow-based Object Rotation Manipulation.** For the arrow-based object rotation manipulation in the augmented reality application, we define this arrow-based interaction using a 2D start point $\mathcal{P}_{start}$ and a 2D end point $\mathcal{P}_{end}$, forming a direction vector $\mathcal{V}_{target} = \mathcal{P}_{end} - \mathcal{P}_{start}$. Then we use Unidepth [31] to estimate the camera intrinsic and depth map. With camera intrinsic, we calculate the 3D camera rays $\mathcal{R}_{start}$ and $\mathcal{R}_{end}$ for the corresponding 2D points $\mathcal{P}_{start}$ and

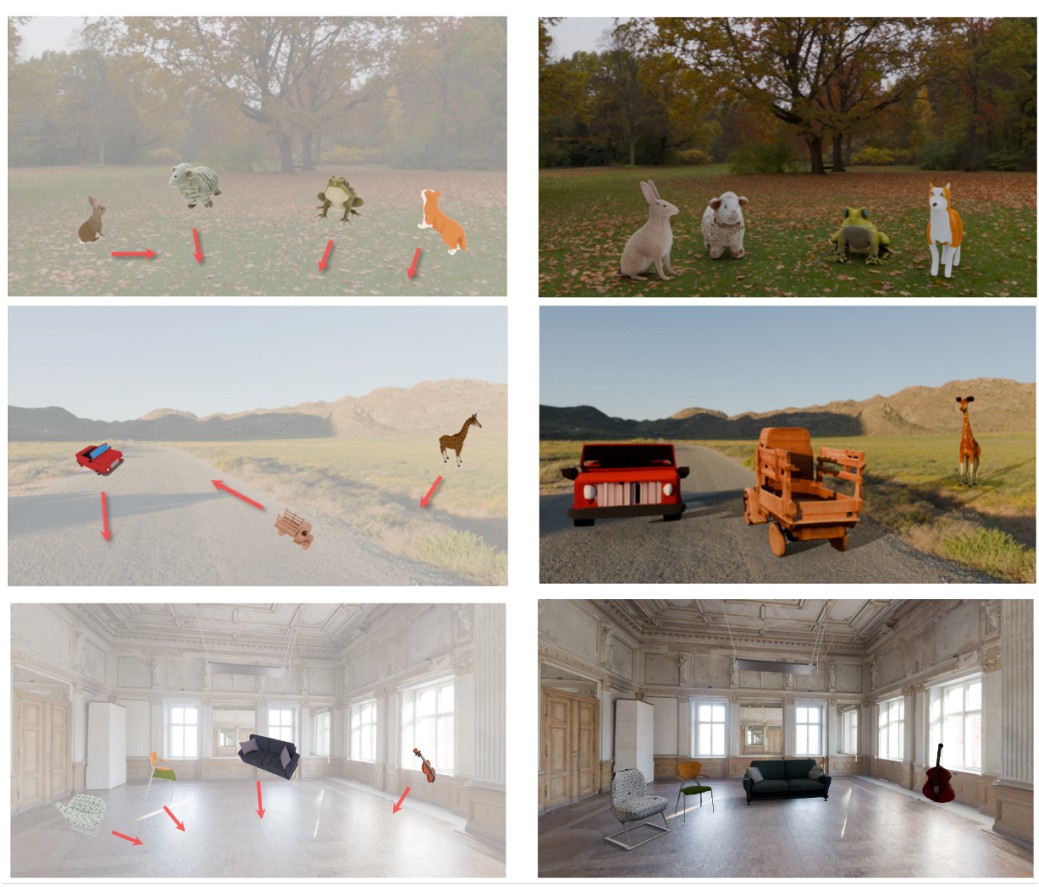

Figure 16: More results of our efficient arrow-based object manipulation method.

$\mathcal{P}_{end}$. With the depth map, we calculate the plane $\mathcal{P}$ via the least square method. After that, we calculate the 3D intersection points $\mathcal{P}^{3d}_{end}$ and $\mathcal{P}^{3d}_{start}$ between the corresponding camera rays $\mathcal{R}_{start}$, $\mathcal{R}_{end}$ and the extracted plane $\mathcal{P}$. Finally, 3D direction vector $\mathcal{V}^{3d}_{target}$ can be calculated: $\mathcal{V}^{3d}_{target} = \mathcal{P}^{3d}_{end} - \mathcal{P}^{3d}_{start}$. Since each generated model is aligned to a known forward vector $\mathcal{V}^{3d}_{init}$, we can directly compute and apply the rotation that aligns $\mathcal{V}^{3d}_{init}$ with $\mathcal{V}^{3d}_{target}$, allowing the object to be placed in the correct orientation without any manual adjustment. Note that the 3D start point $\mathcal{P}^{3d}_{start}$ is also the 3D location of the inserted 3D object, the 3D models need to rotate along $\mathcal{V}^{3d}_{target}$ to ensure verticality to the ground and the object size still needs to be set via VLM or user interaction, since our generated models share a normalized scale. Specifically, to automatically estimate the size of the object to insert, we can directly ask the VLM to estimate the common size of the object in the real world with the single-view image as input, since our method estimates metric depth with the scale aligned with the real world for insertion. Besides, to make the insertion results realistic, we set a plane as a shadow catcher and estimate the environment lighting using DiffusionLight [30]. For the arrow-based object rotation manipulation in the generic 3D software, we define this arrow-based interaction using a 3D start point $\mathcal{P}^{3d}_{start}$ and a 3D end point $\mathcal{P}^{3d}_{end}$, forming a 3D direction vector $\mathcal{V}^{3d}_{target} = \mathcal{P}^{3d}_{end} - \mathcal{P}^{3d}_{start}$. The rotation transformation is calculated by aligning $\mathcal{V}^{3d}_{init}$ with $\mathcal{V}^{3d}_{target}$ using Rodrigues' formula and users can further rotate the objects along $\mathcal{V}^{3d}_{target}$ if needed.

## B  More Results

### B.1  Efficient Arrow-based Object Manipulation

We present more results of our efficient arrow-based object manipulation method in the Figure 16.

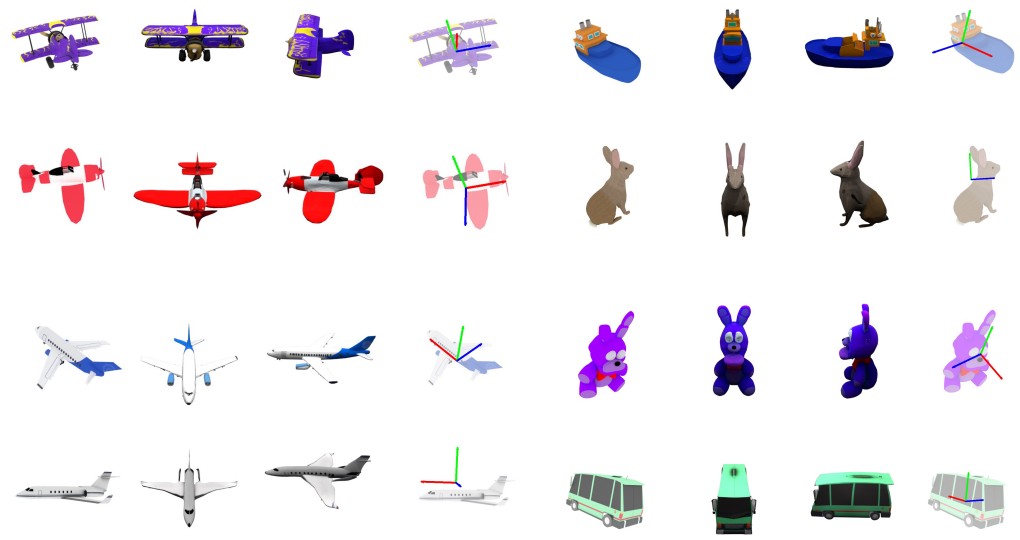

Figure 17: More qualitative results of our Trellis-OA and orientation estimation on Toys4k [39]. Note that the images correspond to the input image, two renderings of the generated 3D model, and the orientation estimation results, respectively.

|  | Chamfer Distance ↓ | LPIPS ↓ | CLIP ↑ |
|---|---|---|---|
| Trellis (Manually corrected) | 0.0377 | 0.1723 | 88.33 |
| Trellis-OA (Ours) | **0.0280** | **0.1574** | **88.77** |

Table 6: Quantitative comparison of geometry and appearance quality between Trellis with manually corrected orientations and Trellis-OA (Ours). The results demonstrate that our orientation-alignment fine-tuning can enhance the performance of the pre-trained checkpoint, likely attributable to the high quality of our Objaverse-OA dataset.

## B.2 Orientation-aligned Object Generation

We present more qualitative results of our Trellis-OA and orientation estimation method in Figure 17, Figure 18, Figure 19, Figure 20, and Figure 21. We also present more qualitative comparisons between Wonder3D and our Wonder3D-OA in Figure 22 and Figure 23. Besides, we present more results of Trellis-OA and Wonder3D-OA in our demo video. Please check the video for details.

To investigate whether our orientation alignment fine-tuning affects the performance of the original pre-trained checkpoint, we randomly select 46 objects across 46 categories in the Toys4k [39] dataset. As shown in Table 6, our method not only maintains performance but may even enhance the quality of the pre-trained 3D generative models. This improvement is likely due to the high fidelity of our manually corrected Objaverse-OA dataset.

## B.3 Orientation Estimation

We detail orientation estimation results of baselines and our method at the category level in Table 7 and Table 8.

## B.4 Impact of Occlusion

As illustrated in Figure 24, our object generation module is capable of producing reasonable results under partial occlusion, with minimal performance degradation. However, in scenarios involving severe occlusion, we recommend using the method proposed in [56], which is specifically designed

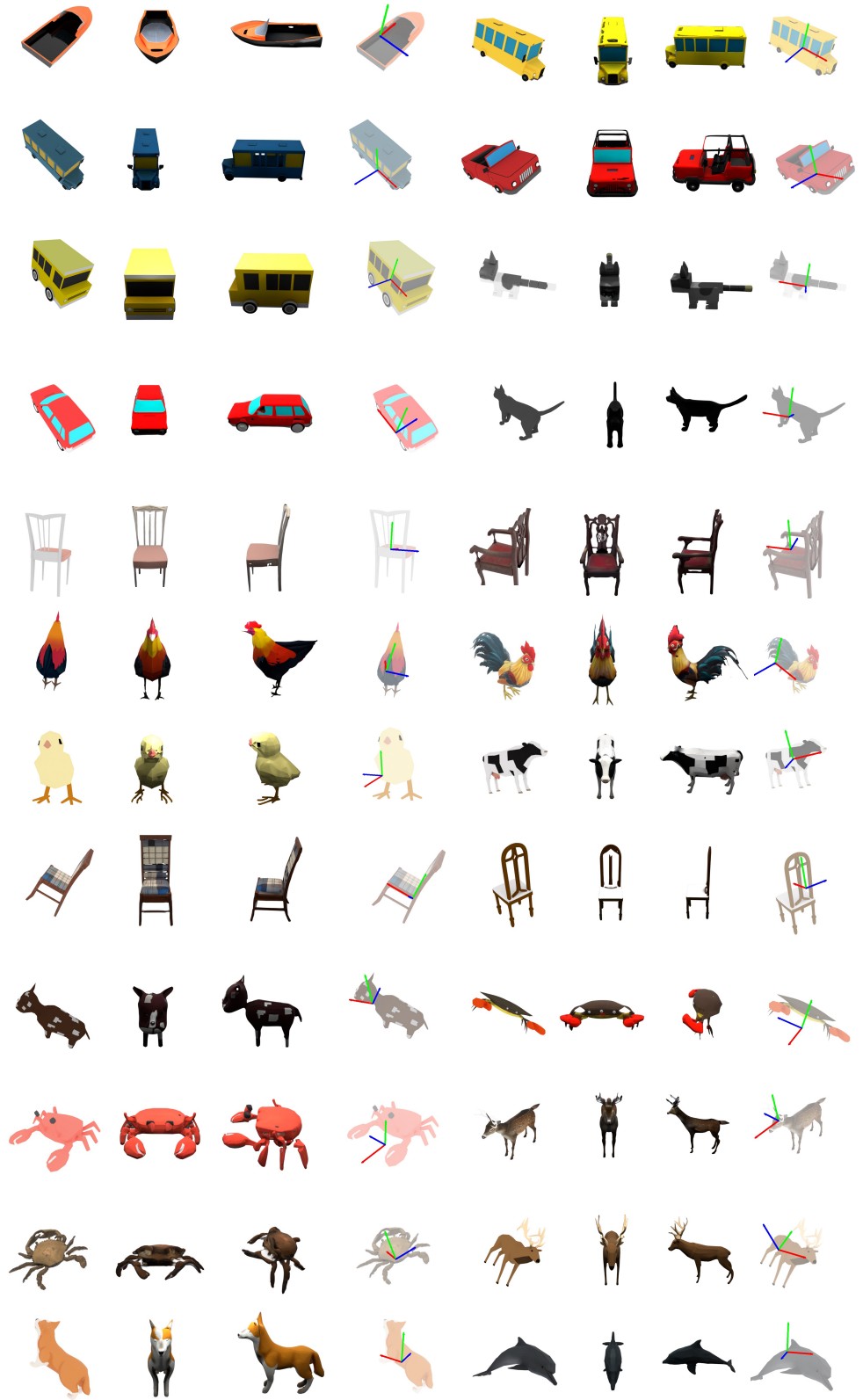

Figure 18: More qualitative results of our Trellis-OA and orientation estimation on Toys4k [39].

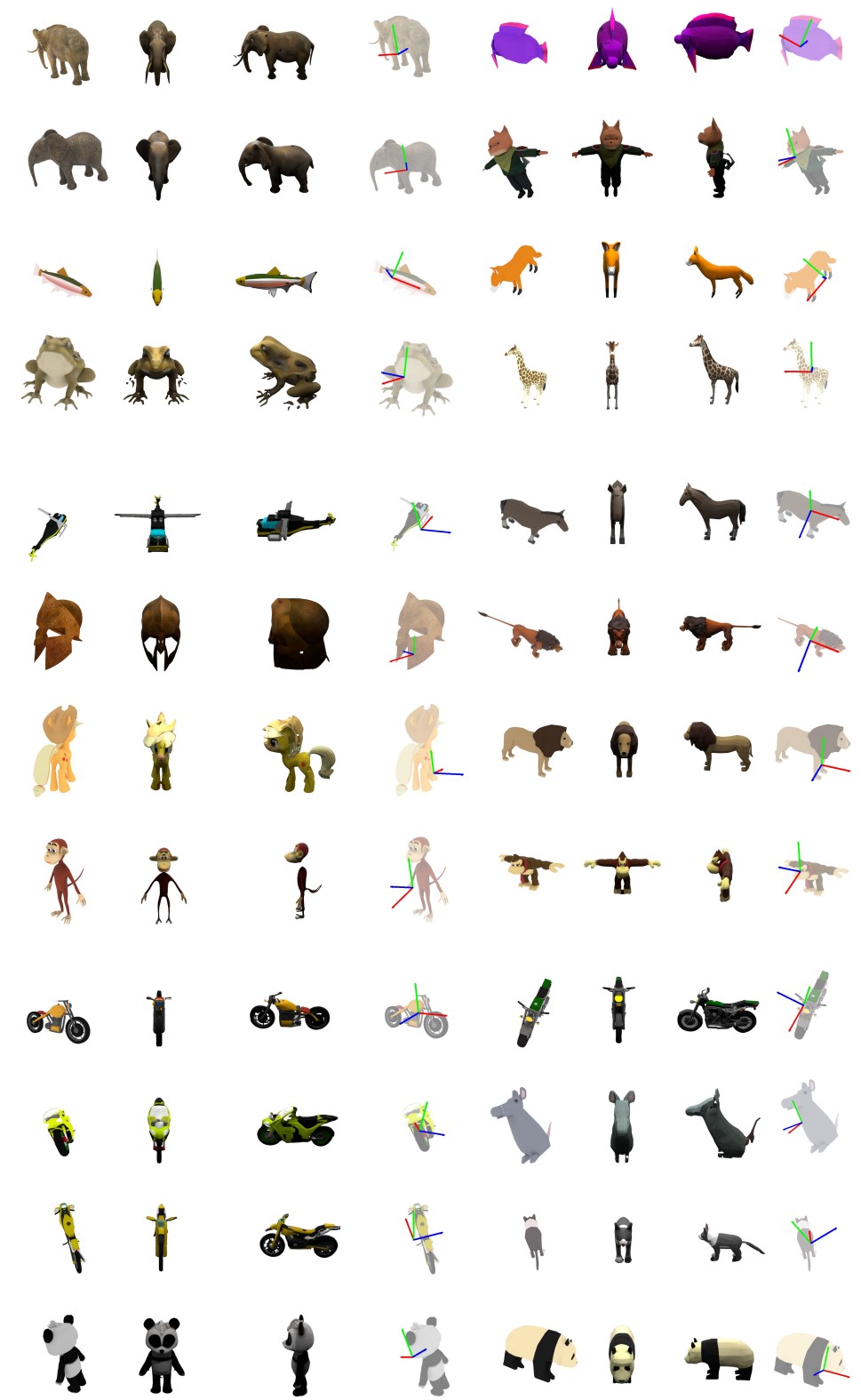

Figure 19: More qualitative results of our Trellis-OA and orientation estimation on Toys4k [39].

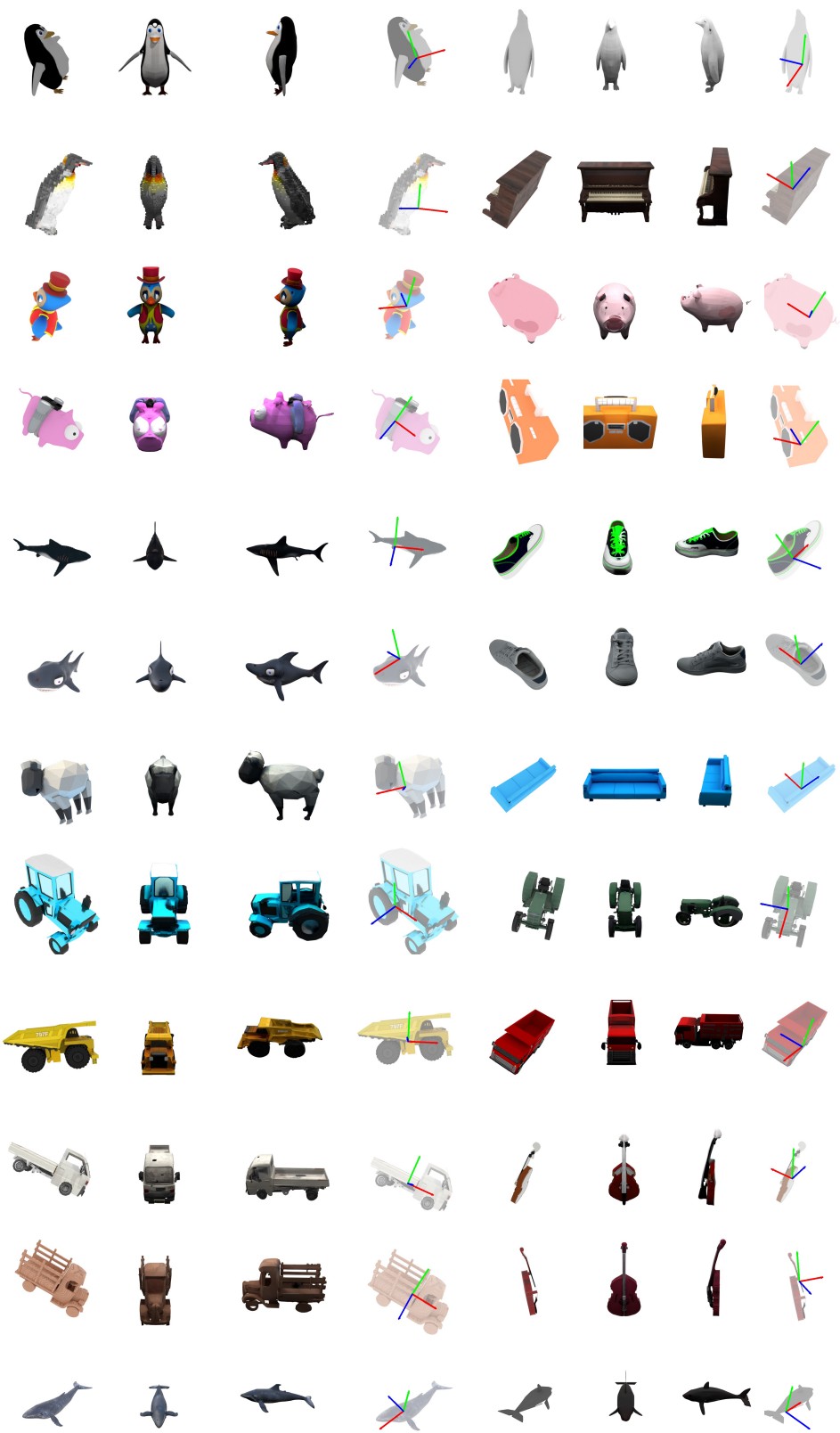

Figure 20: More qualitative results of our Trellis-OA and orientation estimation on Toys4k [39].

|  | airplane | bicycle | boat | bunny | bus | car |
|---|---|---|---|---|---|---|
| FSDetView [58] | - / - | 40.00 / 58.52 | 11.11 / 114.89 | - / - | 20.00 / 75.71 | 40.00 / 58.07 |
| Orient Anything [52] (Vit-S) | 30.00 / 70.48 | 20.00 / 63.91 | 44.44 / 73.69 | 33.33 / 47.79 | 100.00 / 11.64 | 80.00 / 29.49 |
| Orient Anything [52] (Vit-L) | 80.00 / 39.03 | 40.00 / 74.73 | 44.44 / 87.91 | 88.89 / 15.63 | 80.00 / 20.32 | 80.00 / 18.73 |
| Ours (ViT-S) | 66.67 / 55.99 | 80.00 / 40.28 | 55.56 / 66.21 | 44.44 / 50.77 | 100.00 / 9.52 | 60.00 / 74.66 |
| Ours (ViT-L) | 77.78 / 34.96 | 80.00 / 18.76 | 44.44 / 58.47 | 44.44 / 45.00 | 100.00 / 10.55 | 100.00 / 17.15 |
|  | cat | chair | chicken | cow | crab | deer moose |
| FSDetView [58] | - / - | 10.00 / 96.30 | - / - | - / - | - / - | - / - |
| Orient Anything [52] (Vit-S) | 55.56 / 35.79 | 55.00 / 39.50 | 25.00 / 59.75 | 33.33 / 61.73 | 10.00 / 75.44 | 50.00 / 29.89 |
| Orient Anything [52] (Vit-L) | 77.78 / 19.83 | 90.00 / 15.53 | 50.00 / 34.62 | 50.00 / 40.90 | 50.00 / 42.89 | 100.00 / 17.54 |
| Ours (ViT-S) | 22.22 / 54.68 | 45.00 / 59.01 | 37.50 / 73.32 | 33.33 / 41.42 | 40.00 / 69.11 | 50.00 / 30.04 |
| Ours (ViT-L) | 33.33 / 65.48 | 40.00 / 49.99 | 75.00 / 28.66 | 33.33 / 63.09 | 60.00 / 46.70 | 60.00 / 34.98 |
|  | dinosaur | dog | dolphin | dragon | elephant | fish |
| FSDetView [58] | - / - | - / - | - / - | - / - | - / - | - / - |
| Orient Anything [52] (Vit-S) | 50.00 / 32.32 | 22.22 / 53.95 | 30.00 / 86.84 | 20.00 / 69.35 | 50.00 / 45.98 | 20.00 / 59.59 |
| Orient Anything [52] (Vit-L) | 70.00 / 22.11 | 88.89 / 16.71 | 30.00 / 50.37 | 50.00 / 48.06 | 70.00 / 28.76 | 40.00 / 47.36 |
| Ours (ViT-S) | 10.00 / 54.34 | 55.56 / 47.04 | 70.00 / 34.27 | 10.00 / 76.13 | 50.00 / 56.19 | 20.00 / 82.77 |
| Ours (ViT-L) | 40.00 / 38.01 | 44.44 / 46.88 | 60.00 / 34.52 | 20.00 / 59.41 | 70.00 / 35.56 | 20.00 / 89.96 |
|  | fox | frog | giraffe | guitar | helicopter | helmet |
| FSDetView [58] | - / - | - / - | - / - | 0.00 / 110.48 | - / - | 28.57 / 49.01 |
| Orient Anything [52] (Vit-S) | 77.78 / 31.87 | 50.00 / 48.89 | 0.00 / 51.62 | 30.00 / 67.83 | 0.00 / 101.22 | 14.29 / 53.71 |
| Orient Anything [52] (Vit-L) | 66.67 / 22.15 | 90.00 / 21.53 | 80.00 / 24.69 | 50.00 / 35.85 | 50.00 / 74.92 | 14.29 / 54.06 |
| Ours (ViT-S) | 55.56 / 47.99 | 50.00 / 63.22 | 50.00 / 32.03 | 40.00 / 39.72 | 25.00 / 111.61 | 57.14 / 70.63 |
| Ours (ViT-L) | 44.44 / 39.19 | 60.00 / 64.88 | 60.00 / 32.85 | 30.00 / 64.50 | 25.00 / 67.30 | 42.86 / 62.73 |
|  | horse | laptop | lion | lizard | monkey | motorcycle |
| FSDetView [58] | - / - | 62.50 / 42.29 | - / - | - / - | - / - | - / - |
| Orient Anything [52] (Vit-S) | 20.00 / 73.80 | 75.00 / 19.83 | 50.00 / 36.90 | 10.00 / 56.94 | 90.00 / 19.91 | 25.00 / 101.14 |
| Orient Anything [52] (Vit-L) | 80.00 / 36.15 | 50.00 / 70.09 | 70.00 / 28.20 | 40.00 / 38.71 | 70.00 / 21.11 | 50.00 / 51.71 |
| Ours (ViT-S) | 70.00 / 31.55 | 50.00 / 72.41 | 30.00 / 65.78 | 50.00 / 45.07 | 60.00 / 40.70 | 50.00 / 72.01 |
| Ours (ViT-L) | 60.00 / 48.74 | 50.00 / 32.11 | 40.00 / 33.39 | 50.00 / 36.63 | 60.00 / 49.30 | 62.50 / 68.29 |
|  | mouse | panda | PC mouse | penguin | piano | pig |
| FSDetView [58] | 10.00 / 80.55 | - / - | - / - | - / - | 33.33 / 102.13 | - / - |
| Orient Anything [52] (Vit-S) | 20.00 / 43.97 | 57.14 / 30.07 | 0.00 / 116.94 | 50.00 / 49.04 | 50.00 / 64.67 | 30.00 / 50.62 |
| Orient Anything [52] (Vit-L) | 40.00 / 29.71 | 71.43 / 14.63 | 0.00 / 89.58 | 56.25 / 33.90 | 50.00 / 61.28 | 80.00 / 30.68 |
| Ours (ViT-S) | 20.00 / 73.00 | 57.14 / 79.84 | 33.33 / 69.42 | 56.25 / 58.98 | 50.00 / 68.89 | 20.00 / 72.01 |
| Ours (ViT-L) | 40.00 / 51.85 | 57.14 / 55.52 | 0.00 / 74.86 | 31.25 / 57.58 | 50.00 / 56.13 | 40.00 / 67.66 |
|  | radio | robot | shark | sheep | shoe | sofa |
| FSDetView [58] | - / - | - / - | - / - | - / - | 40.00 / 98.08 | 15.00 / 107.20 |
| Orient Anything [52] (Vit-S) | 50.00 / 66.47 | 62.50 / 63.46 | 20.00 / 50.26 | 75.00 / 34.54 | 50.00 / 62.30 | 75.00 / 33.12 |
| Orient Anything [52] (Vit-L) | 50.00 / 23.66 | 87.50 / 32.48 | 40.00 / 45.30 | 100.00 / 16.06 | 50.00 / 32.59 | 65.00 / 39.84 |
| Ours (ViT-S) | 50.00 / 26.10 | 37.50 / 48.90 | 60.00 / 46.41 | 75.00 / 23.73 | 60.00 / 39.92 | 85.00 / 33.35 |
| Ours (ViT-L) | 0.00 / 94.76 | 37.50 / 75.79 | 70.00 / 24.69 | 100.00 / 18.32 | 80.00 / 34.58 | 85.00 / 33.35 |
|  | tractor | train | truck | violin | whale |  |
| FSDetView [58] | - / - | 0.00 / 93.15 | - / - | - / - | - / - |  |
| Orient Anything [52] (Vit-S) | 75.00 / 32.67 | 0.00 / 117.58 | 80.00 / 23.95 | 0.00 / 95.34 | 44.44 / 47.62 |  |
| Orient Anything [52] (Vit-L) | 75.00 / 37.13 | 50.00 / 70.34 | 100.00 / 11.94 | 30.00 / 45.89 | 44.44 / 39.14 |  |
| Ours (ViT-S) | 87.50 / 31.52 | 25.00 / 119.20 | 90.00 / 23.30 | 50.00 / 69.62 | 44.44 / 57.50 |  |
| Ours (ViT-L) | 68.75 / 34.80 | 50.00 / 76.18 | 100.00 / 14.03 | 30.00 / 65.17 | 44.44 / 49.80 |  |

Table 7: Category level quantitative results of orientation estimation on Toys4k [39] dataset in terms of Acc@30 ↑ and Abs ↓.

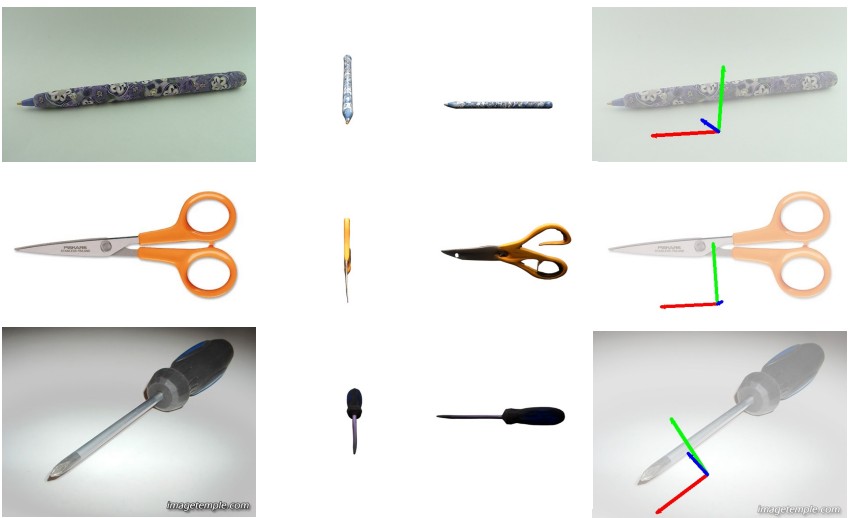

Figure 21: More qualitative results of our Trellis-OA and orientation estimation on Imagenet3D [24] dataset.

|  | fork | knife | pen | rifle | scissors |
|---|---|---|---|---|---|
| FSDetView [58] | 9.09 / 90.67 | 4.76 / 81.40 | 0.00 / 81.72 | 54.84 / 40.24 | 0.00 / 93.31 |
| Orient Anything [52] (Vit-S) | 4.55 / 83.23 | 0.00 / 78.44 | 6.67 / 74.79 | 3.23 / 87.56 | 3.03 / 74.11 |
| Orient Anything [52] (Vit-L) | 9.09 / 78.51 | 0.00 / 78.95 | 13.33 / 76.79 | 6.45 / 81.73 | 3.03 / 85.85 |
| Ours (ViT-S) | 54.55 / 37.43 | 37.50 / 56.61 | 50.00 / 59.02 | 61.29 / 33.89 | 48.48 / 47.15 |
| Ours (ViT-L) | 54.55 / 34.98 | 54.17 / 29.82 | 56.67 / 52.50 | 54.84 / 27.90 | 57.58 / 35.68 |
|  | screwdriver | spoon |  |  |  |
| FSDetView [58] | 0.00 / 96.13 | 0.00 / 103.08 |  |  |  |
| Orient Anything [52] (Vit-S) | 0.00 / 90.53 | 2.63 / 76.98 |  |  |  |
| Orient Anything [52] (Vit-L) | 17.24 / 70.88 | 15.79 / 68.49 |  |  |  |
| Ours (ViT-S) | 79.31 / 21.93 | 78.95 / 26.04 |  |  |  |
| Ours (ViT-L) | 68.97 / 25.18 | 76.32 / 30.10 |  |  |  |

Table 8: Category level quantitative results of orientation estimation on stick-like objects from Imagenet3D [24] dataset in terms of Acc@30 ↑ and Abs ↓.

for occlusion-robustness, as the pre-trained checkpoint prior to applying our orientation-alignment fine-tuning. It is also important to note that our orientation estimation approach has not been designed to handle occluded inputs. Further research is necessary to extend its applicability to occlusion-prone scenarios.

## B.5 Failure Cases

Our method may fail in certain scenarios, as illustrated in Figure 25. For instance, when the input image is captured from the rear of an object, the synthesized 3D model may exhibit suboptimal results in the front view (first row). Additionally, when applied to unseen categories, the generated 3D model may incorrectly incorporate features from other categories present in the training dataset (second row).

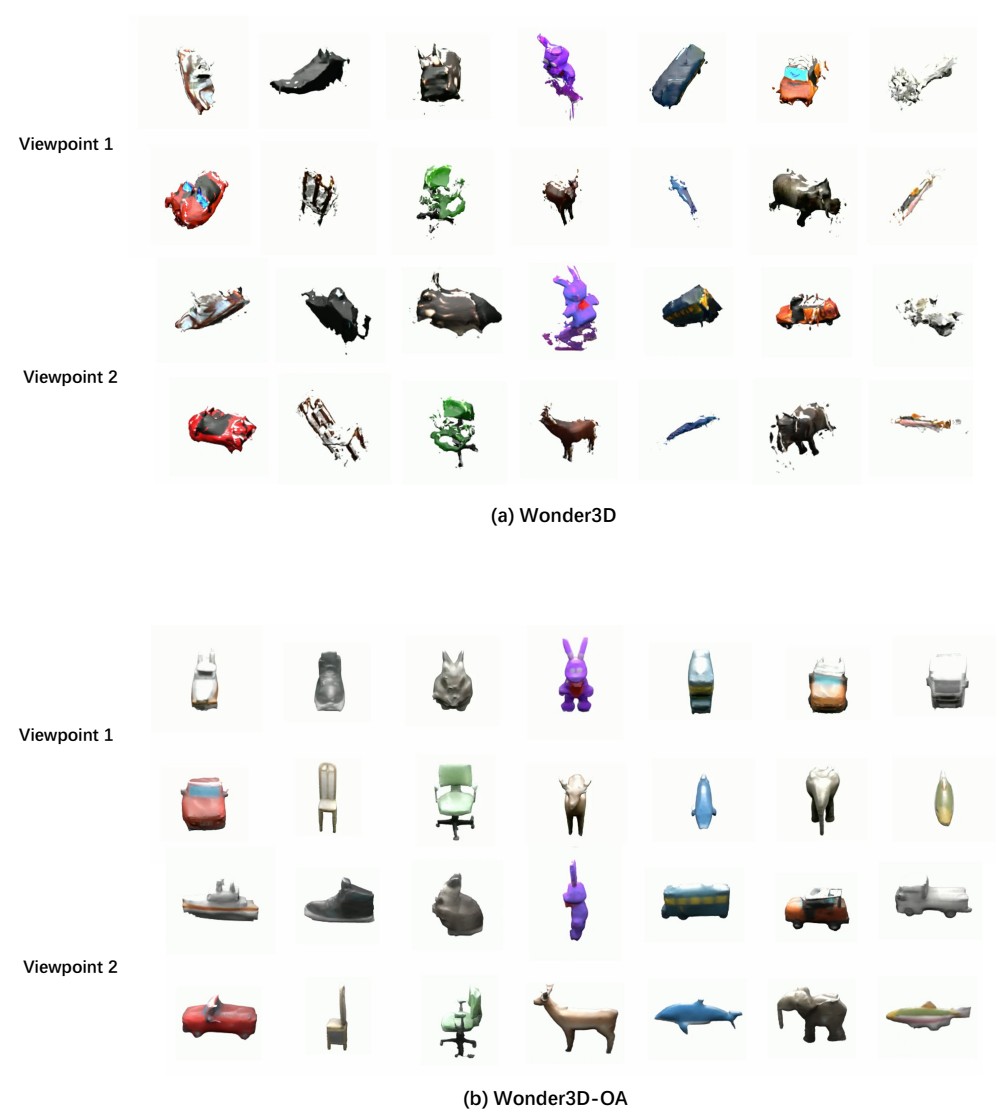

Figure 22: More qualitative comparison between (a) Wonder3D [23] and (b) our Wonder3D-OA.

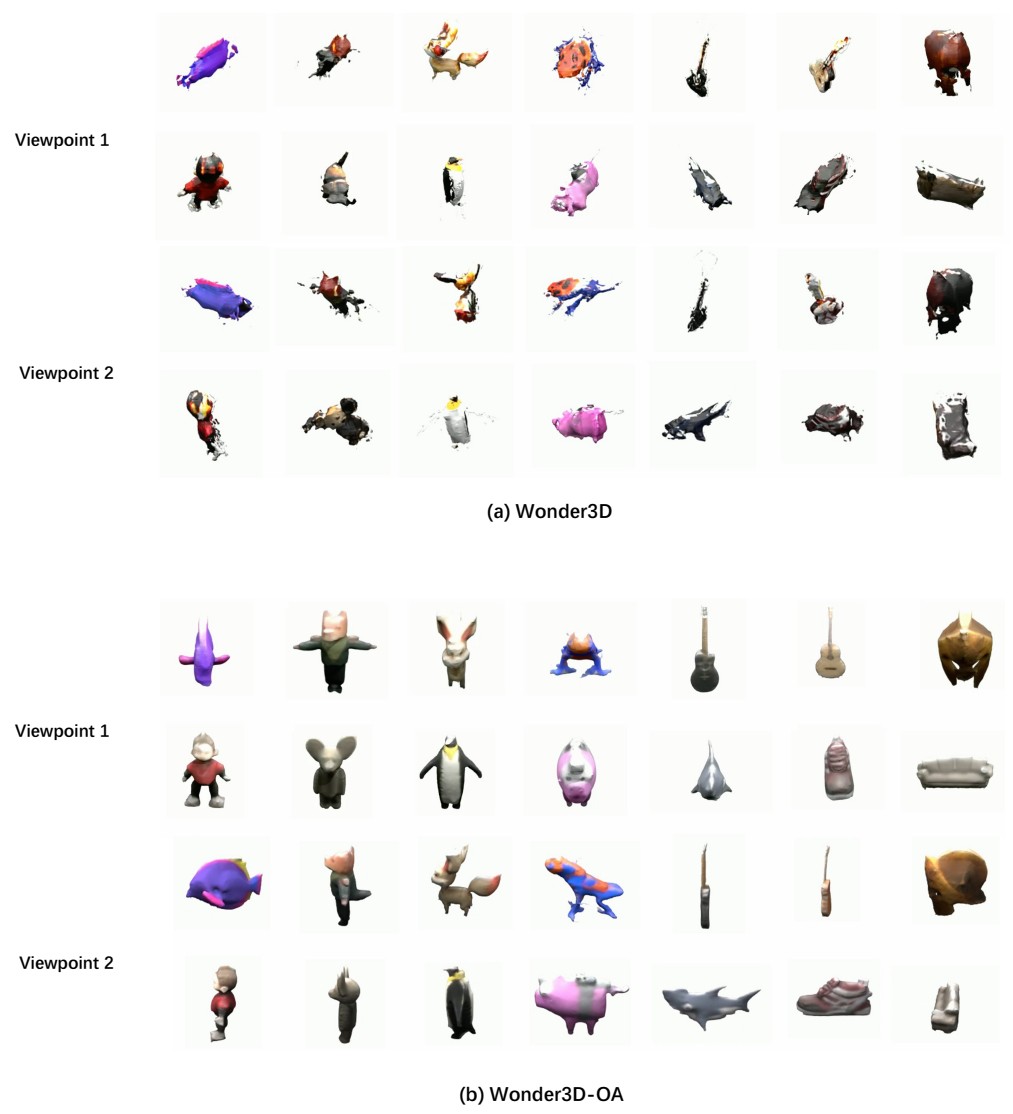

Figure 23: More qualitative comparison between (a) Wonder3D [23] and (b) our Wonder3D-OA.

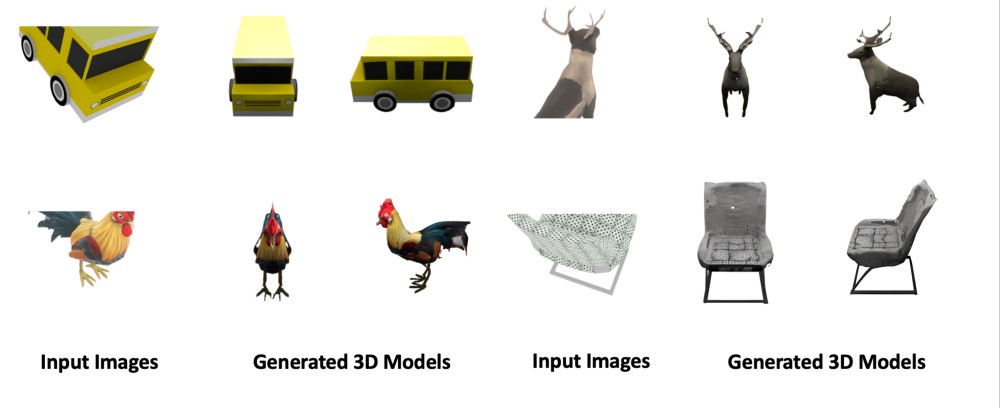

| Input Images | Generated 3D Models | Input Images | Generated 3D Models |

Figure 24: Impact of occlusion

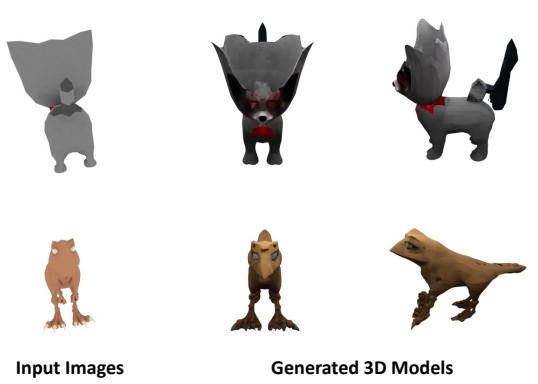

| Input Images | Generated 3D Models |

Figure 25: Failure cases.

## C    License and Border Impact

The licensing details are provided in Table 9. Our method is suitable for generating orientation-aligned 3D objects for downstream applications in 3D perception and augmented reality, which may benefit both scientific research and commercial development. However, it is important to acknowledge potential risks: the method could be misused to generate hazardous 3D objects, such as weapons, which may lead to societal concerns.

| Assets | License | URL |
| --- | --- | --- |
| Blender 4.2.8 | GNU General Public License (GPL) | texthttps://www.blender.org/ |
| Objaverse [6] | ODC-By v1.0 license | texthttps://objaverse.allenai.org/ |
| Trellis [57] | MIT License | texthttps://github.com/microsoft/TRELLIS |
| Wonder3D [23] | MIT License | texthttps://github.com/xxlong0/Wonder3D/tree/main |
| FoundationPose [54] | NVIDIA Source Code License | texthttps://github.com/NVlabs/FoundationPose |
| LGM [44] | MIT License | texthttps://github.com/3DTopia/LGM |

Table 9: License

