# OpenReview forum: "Orientation Matters: Making 3D Generative Models Orientation-Aligned"
_NeurIPS.cc/2025/Conference — NeurIPS 2025 poster_

### Official Review · Reviewer_kr5a · 2025-06-03

**Clarity:** 3
**Significance:** 2
**Originality:** 2
**Rating:** 3
**Confidence:** 3

**Summary:**

In many datasets, objects are often presented in arbitrarily oriented canonical positions, which may not align with human perception of their natural orientation.
This paper addresses the challenge of such non-oriented objects by constructing a large dataset of forward-facing objects, defined as orientations consistent with typical human perception.
The authors train two 3D generative models that, given one or more renderings of an object, predict its corresponding forward-facing orientation.
Additionally, they propose and train two downstream tasks that explicitly involve object orientation.

**Questions:**

1. The paper states that each 3D object is rendered from four horizontal viewpoints—front, back, left, and right.
However, many objects in Objaverse exhibit misalignment or arbitrary canonical orientation, meaning these standard views may not accurately capture the true forward-facing direction. As a result, there's a risk that none of the rendered views actually correspond to the perceptual front of the object.
Furthermore, while it's mentioned that Gemini successfully identifies the front view in ~50% of cases, this raises concerns about the remaining objects—does the method fail because no true front-facing view was included?

2. Did you randomize the input render positions during fine-tuning?

**Ethical Concerns:**

["NO or VERY MINOR ethics concerns only"]

**Final Justification:**

the whole paper is built on making a dataset and train on it
it is not the place for datasets

**Limitations:**

1. Limited Realism: Most examples appear toy-like or low-resolution. It remains unclear whether the proposed approach generalizes well to realistic, lifelike 3D objects.

**Quality:**

2

**Strengths And Weaknesses:**

Strengths:
1. Dataset Creation: Construct a large-scale dataset of forward-facing objects, where each object's orientation aligns with human perception.
2. Model Generalization: Demonstrate that fine-tuning a 3D generative model on this dataset enables generalization across diverse object categories.
3. Qualitative Results: Present high-quality visualizations of correctly oriented objects across diverse categories, showcasing the effectiveness of the approach on Objaverse samples.

Weaknesses
1. Limited Realism: Most examples appear toy-like or low-resolution. It remains unclear whether the proposed approach generalizes well to realistic, lifelike 3D objects.
2. Marginal Novelty: Aside from the construction of a forward-facing 3D object dataset, the work offers limited novelty.
3. Insertion Quality: The insertion results often resemble a simple copy-paste of the forward-aligned object into a scene, lacking evidence of sophisticated integration or context-aware placement. So I can see no special thing done here.

---

> ### Author Rebuttal · Authors · 2025-07-31
>
> Thank you, Reviewer kr5a, for acknowledging our contributions to dataset creation and the strength of our experimental results. Here we aim to address the concerns raised in the review.
>
> **W1.Limited Realism.**
>
> Our model can effectively generate 3D models from realistic, lifelike images. We have presented some qualitative results on the real-world dataset ImageNet3D [1] in the supplementary Figure 7. Unfortunately, due to this year NeurIPS's policy, we cannot include more qualitative results. However, Table 3 and Figure 7 of the main paper demonstrate that our orientation estimation method generalizes well to real-world images from ImageNet3D, indicating our strong performance beyond synthetic data.
>
> As for the realism of the generated 3D models, this aspect is largely dependent on the capability of the underlying 3D generative model. Since our experiments demonstrate that the fine-tuned 3D generative models can still preserve the 3D priors of the backbone, fine-tuning more advanced 3D generative models like Hunyuan3D 2.5 [2] can effectively improve the quality of the generated 3D models.
>
> [1] Ma, Wufei et al. "ImageNet3D: Towards General-Purpose Object-Level 3D Understanding." NeurIPS 2024.
>
> [2] Lai, Zeqiang, et al. "Hunyuan3D 2.5: Towards High-Fidelity 3D Assets Generation with Ultimate Details." Arxiv 2025.
>
> **W2. Marginal Novelty.**
>
> The main contributions of our work are that we are the first to identify the important but overlooked task, orientation-aligned 3D generation, via in-depth research on relevant works in the field of 3D generation. We demonstrate that the most effective solution is creating a new dataset and directly fine-tuning existing 3D generative models rather than commonly used post-processing methods, including PCA, Vision Language Model (VLM), and Orient Anything [3]. We also prove the importance of orientation-aligned 3D generation by firstly applying it to vital downstream applications.
>
> Besides, we also make several other innovations compared to relevant works. For the dataset curation, while the initial stage of our data curation pipeline is inspired by Orient Anything [3], we investigate and address the limitations of the VLM through manual inspection and correction, significantly improving dataset quality. For the orientation estimation application, we are the first to utilize an orientation-aligned 3D generative model for zero-shot model-free orientation estimation. This approach enhances the orientation awareness of SOTA 6D object pose estimation methods like Any6D [4], which use orientation-misaligned generative models for template creation. Our method also exhibits superior generalizability compared to SOTA orientation estimators like Orient Anything [3]. This is because we leverage the comprehensive 3D priors from powerful 3D generative models trained on larger orientation-misaligned datasets, in addition to the rich number of categories in our own dataset. For the object insertion application, we discover that 3D orientation alignment improves the efficiency of object orientation manipulation in simulation systems like the augmented reality application.
>
> [3] Wang, Zehan et al. "Orient Anything: Learning Robust Object Orientation Estimation from Rendering 3D Models." ICML 2025.
>
> [4] Lee, Taeyeop et al. "Any6D: Model-free 6D Pose Estimation of Novel Objects." CVPR 2025.
>
> **W3. Insertion quality and method novelty.**
>
> Our object insertion method primarily focuses on enabling efficient control of 3D object orientation within simulation systems such as augmented reality applications, rather than optimizing for insertion quality. Manually adjusting the orientation of a misaligned 3D model is both time-consuming and labor-intensive, especially in large-scale scenes, which takes approximately 20 seconds for a skilled user to manually rotate a randomly oriented object to the desired orientation or canonical orientation. To address this, we propose an arrow-based object insertion method that allows for intuitive and effective orientation control, made possible by our orientation-aligned 3D generation pipeline.
>
> As described in the Supplementary Material (Line 74), we employ DiffusionLight [5], an environment lighting estimation method, to ensure visual consistency between the inserted objects and the background image. This lighting strategy helps maintain realistic integration without compromising the focus on orientation control. You can check the middle row of the supplementary Figure 2 to see the shadow caused by our environmental lighting.
>
> [5] Phongthawee, Pakkapon et al. "DiffusionLight: Light Probes for Free by Painting a Chrome Ball." CVPR 2024.
>
> **Q1. Reason for the VLM failure.**
>
> As detailed in Supplementary Section A.1, some VLM (Vision-Language Model) failures are indeed attributed to pose misalignment, particularly in the case of stick-like objects where none of the four horizontally sampled viewpoints clearly captures a true front-facing view.
>
> However, as shown by Orient Anything [3], most objects in the Objaverse dataset exhibit primary variation along the horizontal (yaw) axis. To further understand pose misalignment, we conducted an analysis on the Objaverse-LVIS dataset.
>
> We used Principal Component Analysis (PCA) to determine the principal axes of the objects and evaluated whether these axes aligned with the global coordinate system. Due to variations in object shape, some canonical poses yielded uncanonical principal axes under PCA. To mitigate this, we first identified objects in the Objaverse-OA dataset with principal axes aligned to the coordinate system. We then examined the same objects within the Objaverse-LVIS dataset.
>
> The analysis reveals that only 16.83% of objects exhibit arbitrary poses in Objaverse-LVIS. Therefore, while pose misalignment contributes to VLM errors, it is not the sole cause. As we detailed in the supplementary section A.1, the VLM recognition errors can also be caused by other reasons, like unclear front-view feature and ambiguous front-view definition.
>
> **Q2. Randomization of the input rendering positions during fine-tuning.**
>
> Yes, during fine-tuning, we randomly sample camera positions on the unit sphere for rendering input images and also randomize the camera intrinsic, following the official operation adopted by Trellis [6], to ensure the fine-tuned model generalizes well across a diverse range of input views and camera configurations.
>
> [6] Xiang, Jianfeng et al. "Structured 3D Latents for Scalable and Versatile 3D Generation." CVPR 2025.

---

### Official Review · Reviewer_MH83 · 2025-06-22

**Clarity:** 3
**Significance:** 2
**Originality:** 2
**Rating:** 4
**Confidence:** 3

**Summary:**

The paper proposed a novel task on top of image-to-3D generation, i.e. orientation-aligned 3D object generation through data curation and image-to-3D model finetuning (e.g. Wonder3D and Trellis). The proposed method benefits several downstreaming tasks such as pose estimation from a single image and arrow-based object insertion. Extensive experimental results demonstrate the effectiveness of the proposed framework.

**Questions:**

1. How accurate is the proposed method's pose estimation in scenarios involving multiple objects (e.g. some examples shown in orient-anything paper)?
2. In Table 1, why does the vanilla Wonder3D with the normal map branch have a higher Chamfer Distance than Wonder3D-OA? Is this due to the replacement of the 3D representation with LGM, or is it attributed to the orientation-aware design introduced in the paper?

**Ethical Concerns:**

["NO or VERY MINOR ethics concerns only"]

**Final Justification:**

I appreciate the authors' responses, including the extensive experimental results. I would like to raise my score as most of my concerns were faithfully addressed.

**Limitations:**

No potential negative societal impact can be found in this submission. My concern lies primarily in the technical aspects, and I question whether the problem setting itself is truly meaningful.

**Quality:**

3

**Strengths And Weaknesses:**

** Strengths**
1. The methodology and data curation are well-detailed and easy to follow.
2. The experiments are thorough and comprehensive, including both qualitative and quantitative results.
3. The paper implies that an orientation-aligned image-to-3D object generation model benefits single-image pose estimation, which is an interesting insight.

** Weaknesses**
1. The motivation behind the proposed method is not entirely convincing, as 3D orientation does not appear to be a problem in 3D object generation as object pose is typically tied to camera pose during rendering and can be easily adjusted in tools such as Blender. That said, object orientation is arguably more objective than subjective, as it often depends on the inherent functionality of the objects themselves. For example, rendering results of both Wonder3D and Wonder3D-OA are good to me as they are simply rendered from different viewpoints relative to the objects (I know the rendered camera views are actually fixed though).
2. As for object insertion, placing generated objects into a scene layout, such as an indoor room, is typically straightforward with mesh representations. Moreover, relighting the objects along with the scene as a whole is also relatively easy. In contrast, arrow-based object insertion does not seem to support this capability, which may result in inconsistencies in the appearance of inserted objects under novel lighting conditions compared to the original scene layout. Additionally, the global illumination appears to change significantly before and after object insertion in the supplementary video, why is that?

---

> ### Author Rebuttal · Authors · 2025-07-31
>
> Thank you, Reviewer MH83, for acknowledging that our experiments are thorough and comprehensive, and for recognizing that applying our method to single-view pose estimation presents an interesting insight. Here we aim to address the concerns raised in the review.
>
> **W1. The motivation is not entirely convincing.**
>
> We remark that orientation-aligned 3D generation is an important and well-motivated problem, which is also acknowledged by Reviewer yWsi and Reviewer y6ZQ. Next, I will address your concerns one by one.
>
> 1. Object orientation in camera coordinate is also fundamentally influenced by the object's orientation in the world coordinate system, although it is related to the camera pose, since it can be formulated as: $R _ {ob-in-cam} = R _ {ob-in-world} R^{-1} _ {cam-in-world}$, where $R _ {ob-in-cam}$ is the object pose in the camera coordinate system, $R _ {ob-in-world}$ is the object 3D pose in the world coordinate system, and $R _ {cam-in-world}$ is the camera 3D pose in the world coordinate system. Our orientation-aligned 3D generation actually aims to generate 3D models with aligned poses in the world coordinate system. This relationship also explains that the orientation difference in rendering results between Wonder3D and Wonder3D-OA from the same camera pose will lead to a discrepancy in global orientation after lifting them to 3D space.
>
> 2. Manual object pose adjustment in Blender is time-consuming. As you mentioned, to edit the orientation of a 3D model, a commonly used method is manually adjusting its pose in Blender. However, manually rotating a randomly oriented model to its canonical pose or a desired orientation in Blender is time-consuming, taking approximately 20 seconds according to our experiment. For complex scenes with hundreds of objects, this becomes prohibitively inefficient.
>
> 3. Our orientation-aligned 3D generation task is strongly motivated by practical downstream applications, as acknowledged by you and other reviewers. First, 3D models lacking orientation alignment are unsuitable for orientation-related tasks. We demonstrate this with quantitative results for orientation estimation, comparing the usage of orientation-misaligned Trellis to our Trellis-OA during orientation estimation:
>
> |  | ACC@30 on Toys4k $\uparrow$ | Abs on Toys4k $\downarrow$ | ACC@30 on ImageNet3D $\uparrow$ | Abs on ImageNet3D $\downarrow$ |
> |----|:----:|:----:|:----:|:----:|
> | Trellis | 6.65 | 101.94 | 13.73 | 86.49 |
> | Trellis-OA (Ours) | **52.87** | **46.76** | **62.25** | **34.20** |
>
> Second, as mentioned above, misaligned 3D models significantly hinder efficiency in practical use. Manually rotating randomly oriented models is time-consuming, especially for complex scenes. In contrast, our orientation-aligned 3D models offer a more scalable and user-friendly solution, which is highly beneficial for downstream systems, such as the augmented reality applications presented in our paper.
>
> 4. Humans' common sense of object's orientation is already built upon the functionality of the objects: e.g., the front of a television or computer is the viewing direction, while the front of a cabinet is defined by the direction of access. For objects with ambiguous front-view definitions, as also noted by Reviewer y6ZQ (Weakness 1), we adopt the conventions from ImageNet3D [1] for existing categories and follow its principles-semantic part analysis, shape features, and common knowledge-for new categories in our dataset. Please refer to our earlier response to Reviewer y6ZQ for more on this point.
>
> [1] Ma, Wufei et al. "ImageNet3D: Towards General-Purpose Object-Level 3D Understanding." NeurIPS 2024.
>
> **W2. Novelty and illumination problem of the object insertion method.**
>
> While inserting mesh-based 3D objects into a scene and applying relighting techniques may be technically straightforward, controlling object orientation efficiently remains a challenging problem-especially for misaligned models, as previously discussed. Our proposed arrow-based object insertion framework addresses this challenge by providing a simple yet effective mechanism for controlling orientation in augmented reality scenarios. This is made possible due to our orientation-aligned 3D generation pipeline.
>
> Regarding illumination, as stated in Supplementary Material (Line 74), we employ DiffusionLight [2], an environment lighting estimation method, to ensure consistency between inserted objects and the background scene. The global illumination change shown in the supplementary video is solely intended to enhance the visibility of our arrow-based control interface and does not affect the background image itself. Specifically, we add a semi-transparent white background to the background image before insertion to make the input images and arrows more distinct. The background image remains unchanged before and after insertion (the background image after insertion is the background image we use throughout the applications).
>
> [2] Phongthawee, Pakkapon et al. "DiffusionLight: Light Probes for Free by Painting a Chrome Ball." CVPR 2024.
>
> **Q1. Pose estimation in scenarios involving multiple objects.**
>
> To extend our pose estimation method to multi-object scenarios, we can incorporate an object segmentation module, such as SAM [3] or GroundSAM [4], to segment the input scene into individual object regions. These segments are then processed independently by our pose estimation pipeline.
>
> Our method performs well in various multi-object scenarios. However, due to NeurIPS 2025 rebuttal policies, we are unable to include qualitative results for such cases.
>
> Besides, we acknowledge that highly crowded scenes with severe occlusion may still degrade performance. Addressing this issue is part of our planned future work through occlusion-aware training for orientation estimation.
>
> [3] Kirillov, Alexander et al. "Segment Anything." ICCV 2023.
>
> [4] Ren, Tianhe et al. "Grounded SAM: Assembling Open-World Models for Diverse Visual Tasks." ArXiv 2024.
>
> **Q2. The reason why vanilla Wonder3D has a larger Chamfer Distance than Wonder3D-OA.**
>
> To ensure a fair comparison, we clarify that vanilla Wonder3D does not use the normal map during 3D lifting, consistent with Wonder3D-OA. To investigate further, we present additional results that include the use of normal maps with Wonder3D:
>
> |  | Chamfer Distance on GSO $\downarrow$ | Chamfer Distance on Toys4k $\downarrow$ |
> |--|:---:|:---:|
> | Wonder3D w/o normal map | 0.0894 | 0.0932 |
> | Wonder3D w/ normal map | 0.0972 | 0.1018 |
>
>
> Interestingly, the use of normal maps in Wonder3D may even lead to a worse Chamfer Distance. It is because the Chamfer Distance is sensitive to the pose alignment between the evaluated and the GT 3D models. In our experiment, this metric is primarily used to measure pose canonicalization performance across different methods. Since the 3D lifting method may also influence the Chamfer Distance, we replace the LGM [5] with Instant-NGP [6] used for the Wonder3D baseline shown below. The results confirm that the improvements arise predominantly from our orientation-aware design.
>
> |  | Chamfer Distance on GSO $\downarrow$ | Chamfer Distance on Toys4k $\downarrow$ |
> |--|:---:|:---:|
> | Wonder3D-OA w/ Instant-NGP | 0.0609 | 0.0571 |
> | Wonder3D-OA w/ LGM (Ours) | 0.0564 | 0.0548 |
>
> Besides, in our rebuttal to Reviewer y6ZQ W3, we present more quantitative results that only measure the generation quality, disentangled from the pose canonicalization ability, by canonicalizing the 3D models generated by the Wonder3D and Trellis with the GT or manual operation. You can refer to this rebuttal if you have the same concern.
>
> [5] Tang, Jiaxiang et al. "LGM: Large Multi-View Gaussian Model for High-Resolution 3D Content Creation." ECCV 2024.
>
> [6] Müller, Thomas et al. "Instant neural graphics primitives with a multiresolution hash encoding." TOG 2022.

---

### Official Review · Reviewer_y6ZQ · 2025-06-26

**Clarity:** 2
**Significance:** 3
**Originality:** 2
**Rating:** 5
**Confidence:** 4

**Summary:**

The paper first introduces new annotations for a subset of the Objaverse dataset, where objects are orientation-aligned to a pre-defined coordinate system.

Then, two existing 3D generative models, a multi-view generative model, as well as a direct image-to-3D generative model, are fine-tuned using the proposed dataset to generate orientation-aligned 3D objects.

The authors propose estimating the orientation of 3D objects as a downstream application. Using fine-tuned 3D generative models makes an oriented CAD reference model unnecessary. To this end, they adapted and slightly modified a template-based method.
Finally, the authors propose a practical downstream application in augmented/virtual reality scenarios, where object insertion can be steered more easily with the generated coordinate-frame aligned objects as compared to non-aligned objects.

**Questions:**

Please explain the evaluation procedure for computing LPIPS and CLIP scores. Which angles are used? What is the CLIP image embedding compared to? E.g., text prompts ("front view") or the CLIP embedding from the target shape rendering?

The baseline qualitative results look different for the same shape in Fig. 6 (see yellow bus for Trellis + VLM vs. Trellis + Orient Anything). Why? Shouldn't only their rotation change?

In the proposed DINO pose selection stage, measuring feature alignment with the L2 distance should likely be replaced with the cosine similarity. What exactly is compared here? Patch-level or CLS token?

Please feel free to further clarify any misunderstandings given my assessment of strengths and weaknesses of the paper.

**Ethical Concerns:**

["NO or VERY MINOR ethics concerns only"]

**Final Justification:**

The authors addressed my concerns and questions in the rebuttal. While I agree with reviewer yWsi that the practical novelty of the presented work is limited, I believe that the dataset with canonicalized 3D objects will be of help for further research. The paper should be accepted but the authors should update their manuscript following their rebuttal and discussion, especially for the evaluation of multi-view diffusion models.

**Limitations:**

The paper should discuss the ambiguity of orientation annotations for certain object classes more clearly and how this affects evaluation and comparison with prior works.

**Quality:**

2

**Strengths And Weaknesses:**

**Strengths:**

The paper tackles an important problem which is well motivated by the downstream applications shown in the paper, both in robotics and AR/VR applications.
The paper is generally well-written and easy to follow.
The paper introduces new orientation annotations for a subset of the Objaverse dataset, which will be made public and can thus be used in other projects as well.

**Weaknesses:**

The authors state that wrongly annotated orientations by the VLM are manually corrected. As examples, stick-like objects and objects with ambiguous orientation or unclear front view are mentioned.
While "canonicalising" the orientation of ambiguous objects might make sense as a data pre-processing step, it is dangerous to claim that these annotations are "ground truth" from now on.
For example, I am personally unable to tell the "up", "front" or "right" direction for a spoon.

The same problem also is relevant when looking at the comparison against prior work for orientation estimation in Tab.3. The authors argue that their method outperforms prior art for stick-like objects but again it is questionable whether a clear ground truth can be defined in such cases. Maybe, in these cases it would be better to measure whether the principal axes of the object are aligned with any of the coordinate frame axes.

The authors write that "Experiments show that the pre-trained 3D generative models can further improve geometry quality after fine-tuning on our dataset". (l.141)
However, when inspecting the qualitative results in Fig.5, it rather looks like the geometric and visual appearance quality decreases for the generations of the fine-tuned model.

The paper is missing important details on how the experiments are evaluated. To be more specific, the LPIPS (and potentially CLIP) scores need to be computed in comparison against some ground-truth renderings.
If these renderings are e.g. at the six canonical views that were used when training Wonder3D-OA, the comparison in Tab.1 is unfair:
As the LGM (which is used to generate the 3D object) is conditioned on exactly these views for Wonder3D-OA but not for Wonder3D, the renderings at the evaluation viewing angles will include more artefacts caused by the LGM for Wonder3D compared to the OA version.

The presented downstream application (insertion of objects in 3D scenes) using a forward-facing arrow is not taking care of the additional degree of freedom (rotation around the forward facing arrow), which might be desired.
This is, however, only a minor weakness, as the alignment with the coordinate axes still makes manipulation easier than with tilted / rotated objects.

The paper should further include more details on the fine-tuning / introduced changes to prior methods.
E.g., the pixel injector (l.196), not using depth input (l.220), or DINO pose selection stage (l.222) could be explained more clearly.

---

> ### Author Rebuttal · Authors · 2025-07-31
>
> Thank you for recognizing that our work addresses an important problem, well motivated by the downstream applications. Here we aim to address the concerns raised in the review.
>
> **W1. & W2. Ambiguity of object orientation.**
>
> It's true that orientation ambiguity does exist for some objects, especially tools like spoons, mugs, fire extinguishers, and similar items. However, this ambiguity does not prevent the establishment of consistent conventions to define their canonical orientations. It is important to acknowledge that canonical poses may not align perfectly with every individual's intuitive expectations due to inherent ambiguity. However, once a reasonable definition is adopted and agreed upon by the community, it effectively becomes a form of shared common sense. This agreement is crucial for enabling important tasks, such as orientation-aligned 3D generation, object orientation estimation, and efficient object orientation manipulation.
>
> In practice, during the manual correction process, we refer to the object orientations defined in prior work, specifically ImageNet3D [1], for the categories it covers. For example, for spoons, which are included in ImageNet3D, we align their poses in our dataset accordingly. For ambiguous objects only included in our dataset, we define canonical poses based on semantic part structures, geometric features, and common knowledge, following the principles established by ImageNet3D and our supplementary material section A.1.
>
> Due to our pose alignment with ImageNet3D, its human annotation can serve as appropriate ground truth for evaluating our orientation estimation method. Furthermore, relying solely on whether an object's principal axes align with the coordinate frame axes is not a sufficient metric. A model might perform well on such a metric while still failing to distinguish between the front and back of an object. For instance, a robot using an orientation estimation algorithm incapable of identifying the head versus the handle of a spoon would struggle to use it properly.
>
> [1] Ma, Wufei et al. "ImageNet3D: Towards General-Purpose Object-Level 3D Understanding." NeurIPS 2024.
>
> **W3. Qualitative results decrease.**
>
> It is true that the visual geometric and appearance quality of multi-view outputs (Figure 5) exhibits a slight decrease. However, these results are direct outputs of the multi-view diffusion model. The final 3D quality is more strongly influenced by whether the multi-view images are correctly aligned with their corresponding camera views and the 3D lifting method. Since the main paper Table 1 is intended to measure the orientation canonicalization ability, to demonstrate the quality improvement effectively, we canonicalize the poses of Wonder3D with the GT camera poses of the input images and compare its quality to our Wonder3D-OA on the Toys4k dataset:
>
> |  |Chamfer Distance $\downarrow$|LPIPS* $\downarrow$|CLIP* $\uparrow$|
> |-|:-:|:-:|:-:|
> |Wonder3D w/ GT canonicalization|0.0653|0.2460|88.70|
> |Wonder3D-OA (Ours)|0.0548|0.2317|92.09|
>
> Note that the Wonder3D adopts the same multi-view generation and 3D lifting method as we present in the main paper, with only canonicalized with GT poses. The quantitative results demonstrate that our Wonder3D-OA actually has quality improvement over Wonder3D. Note that the LPIPS* and CLIP* are the updated metrics described in the answer to W4 & Q1. Besides, in the supplementary materials (Figure 8, Figure 9, and video 2:18), we also present numerous renderings of the reconstructed 3D models.
>
> Additionally, we also provided similar quantitative results for Trellis-OA in the supplementary materials Table 3. There, we manually rotate the 3D models produced by Trellis and compare the quantitative results to those from our Trellis-OA variant. The results demonstrate that after fine-tuning on our Objaverse-OA, Trellis exhibits better quality performance. Finally, more examples in the supplementary video (e.g., objects shown at 1:42, bottom row; 1:53, row 5, columns 1 and 3; row 6, columns 5, 8, and 10) clearly demonstrate that our Trellis-OA produces higher-quality outputs compared to the original Trellis model.
>
> **W4 & Q1. Details and unfairness in experiment evaluation.**
>
> In our experiments, we originally rendered four orthogonal views with the camera elevation angle of 0 for both the generated and GT 3D models, and computed the LPIPS and CLIP scores based on these renderings at matched camera poses. We acknowledge that this setup may introduce bias, as it aligns with the multi-view configurations used by Wonder3D-OA. To ensure a fairer evaluation, we have updated the rendering protocol by randomly sampling views on a unit sphere for all evaluation objects. We then computed updated metrics denoted as LPIPS* and CLIP* under this revised setup:
>
> |  | LPIPS* on GSO $\downarrow$ | CLIP* on GSO $\uparrow$ | LPIPS* on Toys4k $\downarrow$ | CLIP* on Toys4k $\\uparrow$ |
> |-|:-:|:-:|:-:|:-:|
> | Wonder3D | 0.2799 | 76.37 | 0.2859 | 87.10 |
> | Wonder3D + PCA | 0.2554 | 77.80 | 0.2691 | 87.58 |
> | Wonder3D + Orient Anything | 0.2600 | 77.50 | 0.2699 | 88.12 |
> | Wonder3D + VLM (Gemini-2.0) | 0.2752 | 76.30 | 0.2804 | 87.53 |
> | Wonder3D-OA (Ours) | **0.2270** | **80.30** | **0.2317** | **92.09** |
>
> These updated results confirm that our method continues to outperform the baselines. We will update the results in our final version.
>
> In terms of the reconstruction method applied to the generated images, we used the Instant-NGP [2] adopted in the Wonder3D official implementation to generate the 3D models for all results in the main paper, as we shared the same concern that LGM [3] may be more compatible with Wonder3D-OA than with the original Wonder3D framework.
>
> There is another potential question mentioned by Reviewer MH83 that you may also have: what actually lead to the quality differences? The replacement of the 3D representation with LGM or the orientation-aware design? To address this concern, we further make an ablation study on the LGM design, where we replace the LGM with the Instant-NGP:
>
> |  |CD on GSO $\downarrow$|LPIPS* on GSO $\downarrow$|CLIP* on GSO $\uparrow$|CD on Toys4k $\downarrow$|LPIPS* on Toys4k $\downarrow$|CLIP* on Toys4k $\uparrow$|
> |-|:-:| :-:| :-:| :-:|:-:|:-:|
> | Wonder3D w/ Instant-NGP | 0.0894 | 0.2799 | 76.37 | 0.0932 | 0.2859 | 87.10 |
> | Wonder3D-OA w/ Instant-NGP | 0.0609 | 0.2300 | 80.22 | 0.0571 | 0.2351 | 91.33 |
> | Wonder3D-OA w/ LGM (Ours) | 0.0564 | 0.2270 | 80.30 | 0.0548 | 0.2317 | 92.09|
>
> These results indicate that while LGM [3] contributes to improvements, the primary performance gains over the baselines in the main paper Table 1 are attributed to our orientation-aligned design.
>
> [2] Müller, Thomas et al. "Instant neural graphics primitives with a multiresolution hash encoding." TOG 2022.
>
> [3] Tang, Jiaxiang et al. "LGM: Large Multi-View Gaussian Model for High-Resolution 3D Content Creation." ECCV 2024.
>
> **W5. Without taking care of the additional degree of freedom during object insertion.**
>
> We agree that using a forward-facing arrow alone does not account for the rotational degree of freedom around that axis. However, most real-world objects have a well-defined "up" axis due to their typical placement, which resolves this ambiguity. That said, certain objects, especially hand-held objects, actually lack a clearly defined vertical orientation. For such cases, our pipeline can be further extended to support rotational freedom around the forward-facing axis. Even with this flexibility, our arrow-based method still facilitates more efficient and effective object placement, as you rightly noted.
>
> **W6 & Q3. More details of the introduced changes to prior methods.**
>
> Thanks for pointing out the lack of some details of our method. We will add the following details in our final version:
>
> For the pixel injector (l.196), we follow the design in ImageDream [4] by concatenating the input image with the outputted noisy images for self-attention during training and inference to preserve the local feature in the input image. Specifically, Wonder3D employs a 3D dense self-attention mechanism with a shape of ($b _ z$, 6, $c$, $h _ l$, $w _ l$) across six views within a transformer layer, where $b _ z$ is the batch size, $c$ is the number of feature channels, $h _ l$ and $w _ l$ are the image resolution. Our pixel controller modifies this to ($b _ z$, 7, $c$, $h _ l$, $w _ l$), incorporating the input image as an additional view.
>
> For not using depth input during orientation estimation (l.220), we replace the depth map input of FoundationPose [5] with an all-zero tensor of the same shape, thereby removing any reliance on depth input.
>
> For the DINO pose selection stage (l.222), we obtain the patch feature maps from DINOv2 [6] for both input images and renderings of generated 3D models. Because the best-matching rendering should closely align with the input in both feature directions and norms, we use L2 distance (rather than cosine similarity), as it captures differences in both feature direction and magnitude.
>
> [4] Wang, Peng and Yichun Shi. "ImageDream: Image-Prompt Multi-view Diffusion for 3D Generation." ArXiv 2023.
>
> [5] Wen, Bowen et al. "FoundationPose: Unified 6D Pose Estimation and Tracking of Novel Objects." CVPR 2024.
>
> [6] Oquab, Maxime et al. "DINOv2: Learning Robust Visual Features without Supervision." ArXiv 2023.
>
> **Q2. Difference in qualitative results of two baselines.**
>
> The yellow buses for Trellis + VLM vs. Trellis + Orient Anything in Fig.6 are actually the same 3D model posed in different orientations. The perceived differences likely arise from lighting variations, as we use 3 point lights for illumination following the Trellis [7] official implementation. Variations in object orientation result in differing shadow and highlight patterns, which can create the appearance differences.
>
> [7] Xiang, Jianfeng et al. "Structured 3D Latents for Scalable and Versatile 3D Generation." CVPR 2025.

---

> > ### Comment · Reviewer_y6ZQ · 2025-08-01
> >
> > Thank you for your clarifications.
> >
> > I didn't see my question regarding the CLIP score and how it is computed answered. Could you please explain this?
> >
> > I am quite curious why the CLIP and LPIPS scores seem to have improved with evaluation from "novel" views compared to the diffusion model-generated viewpoints. Do you have any explanation for this or is the evaluation inconsistent with the one used in the paper?
> >
> > I agree with your argument that "once a reasonable definition is adopted and agreed upon by the community, it effectively becomes a form of shared common sense." However, my point was related to the comparison of zero-shot orientation estimation in the paper. OrientAnything (or also Gemini) are not aware of this *canonicalization* step which you introduced through your annotations and are, as myself as well, not capable to tell the top or front of a spoon etc. For this, I think it would be more fair to compare only something like alignment of principal object axes with the coordinate system axes in Tab.3 (for stick-like objects).

---

> > > ### Author Response · Authors · 2025-08-04
> > > **Looking forward to follow-up discussions!**
> > >
> > > Thank you, Reviewer y6ZQ, for taking precious time to check our responses. We hope our answers and additional results address your concerns well. Specifically,
> > >
> > > - Q1: We have clarified the implementation details of the CLIP score.
> > > - Q2: We have explained the differences in the updated LPIPS* and CLIP* results.
> > > - Q3: We have clarified our evaluation of the stick-like objects.
> > >
> > > Please let us know if you have any additional or follow-up questions. We will be more than happy to clarify them. Any follow-up discussions are highly appreciated!

---

> ### Author Response · Authors · 2025-08-02
> **Reply to Official Comment by Reviewer y6ZQ**
>
> Thank you for your quick reply! Here we aim to address your remaining concerns.
>
> **Q1. Clarification on the CLIP score.**
>
> We have provided an overview of our CLIP score implementation in our response to W4 & Q1. To elaborate further, we begin by rendering both the generated 3D models and the GT 3D models from four camera poses, respectively. For the original CLIP score, these camera poses are fixed, specifically, a polar angle of 90° and azimuth angles of 0°, 90°, 180°, and 270°, corresponding to the front, right, back, and left views, respectively. In contrast, our updated CLIP* score uses four randomly sampled viewpoints from the unit sphere. Next, we extract image embeddings using the CLIP image encoder, as mentioned in your Question 1. Finally, we compute the cosine similarity between the embeddings of the generated and GT models under corresponding camera poses. The final CLIP score is obtained by averaging these cosine similarities across all views.
>
> **Q2. Explanation of the differences in the updated evaluation results.**
>
> We appreciate your thoughtful question. The implementations of LPIPS* and CLIP* are consistent with those used in the main paper, apart from the camera poses used for rendering. Regarding the LPIPS*, we would like to remark that the updated results exhibit only minor variations across all methods on both the GSO and Toys4k datasets. This result suggests that the LPIPS metric is not significantly affected by the previously mentioned potential unfairness.
>
> In contrast, the updated CLIP* results indeed show consistent improvements for **all baselines** and Wonder3D-OA. To better understand this phenomenon, we further analyzed the behavior of the original CLIP score and CLIP* on a few representative objects. We observe that the original CLIP score, which is evaluated on front, right, back, and left views, tends to drop significantly when the evaluated rendering is replaced with one captured from a non-corresponding camera pose. In contrast, CLIP* demonstrates only a slight degradation under similar pose mismatches.
>
> We hypothesize that this is due to the nature of CLIP's latent space, which is more sensitive to orientation differences when canonical views are involved, compared to the random views. This sensitivity leads to a lower score in the original CLIP metric. This phenomenon likely stems from the CLIP encoders' pre-training process. CLIP is trained to align text with images, and front views of objects are typically more descriptive and textually grounded. As a result, embeddings of front-view renderings differ more sharply from misaligned ones. On the other hand, random views are harder to describe with text, making their CLIP embeddings less sensitive to pose variation—even when the underlying viewpoint difference is large.
>
> Importantly, regardless of which metric is used, original LPIPS/CLIP or updated LPIPS*/CLIP*, Wonder3D-OA consistently outperforms all Wonder3D-based baselines. This consistency reinforces the robustness and effectiveness of our method under both evaluation protocols.
>
> **Q3. Evaluation of the stick-like objects.**
>
> We agree that for ambiguous objects, misalignment between the canonicalization and common-sense orientation could potentially introduce bias in evaluation, particularly for methods like Orient Anything [1] or Gemini, which are unaware of the canonicalization step. However, for all stick-like objects considered in our paper, the front-view axis is defined in accordance with common-sense orientation. Specifically, the handle is treated as the negative axis and the head as the positive.
>
> Furthermore, we have accounted for directional ambiguity in the other two axes, as stated in our main paper (line 260), where we clarified that we measured only the alignment of the front-view direction for the stick-like objects. This front-view definition is visualized using a red arrow in Figure 7 (bottom row) of the main paper and the right column of Figure 7 in the supplementary materials. These visualizations confirm that our front-view definition is well aligned with common-sense expectations. Note that for other objects without orientation ambiguity, we followed NOCS [2] to calculate the rotation error across all three axes, as we stated in the main paper (line 259).
>
> Therefore, we believe our evaluation remains both fair and more effective than those based on principal axes alignment, as discussed in our response to W1 & W2.
>
> [1] Wang, Zehan et al. "Orient Anything: Learning Robust Object Orientation Estimation from Rendering 3D Models." ICML 2025.
>
> [2] Wang, He et al. "Normalized Object Coordinate Space for Category-Level 6D Object Pose and Size Estimation." CVPR 2019.
>
> Please feel free to let us know if we have misunderstood any part of your question. We would appreciate any further feedback you may have.

---

> > ### Comment · Reviewer_MH83 · 2025-08-04
> > **thanks**
> >
> > I appreciate the authors' detailed responses with additional experimental results. And my concerns w.r.t. the authors' responses are as below:\
> > To authors' responses of **W1**:\
> > I acknowledge the relationship between the rendering result, camera pose and object pose itself in world coordinate space. My primary concern is the way or definition of measuring the alignment. Specifically, the paper measures the object's pose in camera space in $R_{obj-in-cam}$ while keeping $R_{cam-in-world}$ unchanged. However, the world space we are referring to is just a canonical space but no physical (e.g. metric scale and orientation) meaning, which means that the alignment exists only between object sand this canonical space.  This implies  the alignment will likely become invalid if that space is no longer the same canonical space. For example, the arrow-based object insertion with the proposed method will likely fail when the implicit scene layout in 3D space is not in a canonical space (e.g. up to a rotation). \
> > To authors' responses of **W2**:\
> > I see, authors are expected to be more transparent regarding the details of the demo either in the paper or appendix as such enhancement appears quite prominent.  \
> > To authors' responses of **Q1**:\
> > I see, that makes sense to me. \
> > To authors' responses of **Q2**:\
> > I see, the settings should be detailed more clearly in the paper. Also, why the use of normal maps in Wonder3D might lead to a worse Chamfer Distance when supervised with GT 3D models. Are the GT models not well-aligned with the canonical world coordinate space (I don't think so)?
> >
> > Besides, I agree that manual alignment is time consuming and error-prone, but my point is that the definition of a canonical space can be flexible and there seems no point to expect a special model design to get "axis-aligned" 3D models as it's both easy and equivalent to simply rotate and translate 3D models in software like Blender.

---

> > > ### Author Response · Authors · 2025-08-05
> > > **Reply to Official Comment by Reviewer MH83**
> > >
> > > Thank you, Reviewer MH83, for your thoughtful and constructive feedback. We greatly appreciate your engagement with our work. Below, we address your remaining concerns in detail:
> > >
> > > **W1. Canonical space definition.**
> > >
> > > We would like to clarify a potential misunderstanding regarding our definition of "canonical space" one by one.
> > >
> > > > the paper measures the object's pose in camera space in $R _ {ob-in-cam}$ while keeping $R _ {cam-in-world}$ unchanged
> > >
> > > We believe there may be a misunderstanding. Referring to the equation in our initial rebuttal $ R _ {ob-in-cam} = R _ {ob-in-world} R^{-1} _ {cam-in-world} $, our goal is just to clarify that $R _ {ob-in-world}$ is consistent across different object categories. This does **not** imply that $R _ {cam-in-world}$ is unchanged. Since $R _ {ob-in-cam}$ varies across different cameras, $R^{-1}_{cam-in-world}$ changes correspondingly given the fixed $R _ {ob-in-world}$.
> > >
> > > > the world space we are referring to is just a canonical space but no physical (e.g. metric scale and orientation) meaning, which means that the alignment exists only between objects and this canonical space.
> > >
> > > If we understand correctly, the concern is whether the world coordinate system must be aligned with our canonical object coordinate system. We would like to clarify that:
> > >
> > > - The **world coordinate system** can be arbitrarily defined. For example, its axes could be aligned with the camera, the room, or even the sky. It does not need to be aligned with our canonical object coordinate system.
> > >
> > > - Our **canonical object coordinate system** has orientation aligned to common-sense conventions (e.g., the positive x-axis points to the front-view of the object, the positive z-axis points to the up-view of the object), which has the “physical meaning” as you noted.
> > >
> > > No matter how the world coordinate system is defined, once it is determined, the $R _ {ob-in-world}$ is fixed and is consistent across different object categories.
> > >
> > > > This implies the alignment will likely become invalid if that space is no longer the same canonical space.
> > >
> > > Since our canonical space has "physical meaning", this canonical space is actually fixed, unlike the flexibility of the world coordinate system. If you were referring to the intrinsic ambiguity in the object orientation, we have discussed this in our rebuttal to Reviewer y6ZQ's W1 & W2. You can refer to this rebuttal for details.
> > >
> > > > For example, the arrow-based object insertion with the proposed method will likely fail when the implicit scene layout in 3D space is not in a canonical space (e.g. up to a rotation).
> > >
> > > We interpreted your concern as the world coordinate system of the scene we insert objects needs to be aligned with the canonical object coordinate system (please correct us if we misinterpreted your concern). This is not the case.
> > >
> > > As described in the main paper (line 236) and in Supplementary Section A.3, we first estimate the ground plane of the scene, and the inserted object is then transformed to lie on this plane. The forward-facing direction is specified by a user-provided arrow. Thus, regardless of how the scene’s world coordinates are defined, we are able to transform from the fixed canonical object space to the scene appropriately.
> > >
> > > > Besides, I agree that manual alignment is time consuming and error-prone, but my point is that the definition of a canonical space can be flexible and there seems no point to expect a special model design to get "axis-aligned" 3D models as it's both easy and equivalent to simply rotate and translate 3D models in software like Blender.
> > >
> > > We respectfully disagree. As we discussed above, our **canonical object coordinate system** is actually fixed and has “physical meaning”. Therefore, it is sensible to generate “axis-aligned” 3D models with orientation aligned with human expectations. While manual rotation in Blender is technically possible, as you rightly noted, it is also time-consuming and error-prone, particularly at scale. In contrast, our orientation-aligned 3D models support more efficient arrow-based rotation manipulation.
> > >
> > > **W2 & Q1. Demo details and scenarios involving multiple objects.**
> > >
> > > Thanks for your suggestions! We will add more details of the demo and more qualitative results on scenarios involving multiple objects in our final version.
> > >
> > > **Q2. The reason why using normal maps might lead to a worse Chamfer Distance.**
> > >
> > > We appreciate your attention to this point and will clarify this aspect in the final version. The increased Chamfer Distance when using normal maps stems from the difference in coordinate systems between the generated and ground-truth models.
> > >
> > > Specifically, Wonder3D adopts an **input-image related coordinate system**, which means the reconstructed objects' poses are not aligned, as we illustrate in the supp video (02:20). However, the GT models are aligned with the **canonical object coordinate system**. As a result, the Chamfer Distance may increase despite visually plausible reconstructions.

---

> > > ### Author Response · Authors · 2025-08-06
> > > **Further Clarification on Our Canonical Space**
> > >
> > > Dear Reviewer MH83,
> > >
> > > In case there is still any remaining confusion regarding our definition of canonical space, we would like to point you to several related works that adopt similar concepts of orientation-aligned canonical space. Specifically, NOCS [1], ImageNet3D [2], and Orient Anything [3] all aim to establish consistent canonical coordinate systems for objects.
> > >
> > > Our work builds upon and extends this line of research by broadening the concept of canonical space to a wider range of object categories and integrating it into the context of 3D generative models, which has been relatively underexplored.
> > >
> > > We hope this clarifies the motivation and positioning of our work within the broader literature.
> > >
> > > [1] Wang, He et al. “Normalized Object Coordinate Space for Category-Level 6D Object Pose and Size Estimation.” CVPR 2019.
> > >
> > > [2] Ma, Wufei et al. “ImageNet3D: Towards General-Purpose Object-Level 3D Understanding.” NeurIPS 2024.
> > >
> > > [3] Wang, Zehan et al. “Orient Anything: Learning Robust Object Orientation Estimation from Rendering 3D Models.” ICML 2025.
> > >
> > > Please feel free to let us know if we have misunderstood any part of your question. We would appreciate any further feedback you may have.

---

### Official Review · Reviewer_yWsi · 2025-07-01

**Clarity:** 4
**Significance:** 4
**Originality:** 2
**Rating:** 5
**Confidence:** 4

**Summary:**

This paper tackles the challenge of inconsistent orientation in 3D generative models by introducing Objaverse-OA, a large-scale dataset of 14,832 orientation-aligned 3D models across 1,008 categories. Leveraging this dataset, the authors fine-tune two existing 3D generative models (Trellis and Wonder3D) to produce orientation-aligned outputs. The aligned models enable two compelling downstream applications: zero-shot object orientation estimation and efficient arrow-based object insertion.

**Questions:**

1. I suggest that the authors include some evaluation for their dataset construction
pipeline with more metrics (like GEC/GC, ICC), as done in [2], [3], and [5], using a
dataset with ground truth like ShapeNet [4] or similar datasets.
2. A concurrent work that created the Canonical Objaverse Dataset [5] is worthy of
mention if the paper is accepted, but you do not need to compare to it in this
submission.

[2] Rohith Agaram, Shaurya Dewan, Rahul Sajnani, Adrien Poulenard, Madhava Krishna,
Srinath Sridhar, Canonical Fields: Self-Supervised Learning of Pose-Canonicalized Neural
Fields, CVPR 2023
[3] Rahul Sajnani, Adrien Poulenard, Jivitesh Jain, Radhika Dua, Leonidas J. Guibas,
Srinath Sridhar, ConDor: Self-Supervised Canonicalization of 3D Pose for Partial Shapes,
CVPR 2022
[4] Angel X. Chang et al., ShapeNet: An Information-Rich 3D Model Repository
[5] Jin, Li and Wang, Yujie and Chen, Wenzheng and Dai, Qiyu and Gao, Qingzhe and
Qin, Xueying and Chen, Baoquan, One-shot 3D Object Canonicalization based on
Geometric and Semantic Consistency, CVPR 2025

**Ethical Concerns:**

["NO or VERY MINOR ethics concerns only"]

**Final Justification:**

The rebuttal has largely addressed my concerns. While I'm still not 100% concinced about the level of novelty, I think this paper still deserves to be accepted.

**Limitations:**

Not really.
While Section 7's title is "Conclusion and Limitation", this section hardly touches limitations.

**Paper Formatting Concerns:**

No concerns here

**Quality:**

4

**Strengths And Weaknesses:**

Strengths:
1. The paper is well written.
2. Making 3D Generative Models Orientation-Aligned is a nice and important task.
3. I appreciate the significant amount of work presented here: constructing the data, tackling an important problem for the first time, providing nice downstream tasks, creating baselines and a user-friendly interface for object insertion, as well as great figures, supplementary material, etc.

Weaknesses:
I have a hard time deciding whether there are significant theoretical/algorithmic contributions. Essentially, the authors used the exact same method and pipeline from Orient Anything [1] to annotate a large-scale dataset (which is valuable but not novel, and this paper is not in the NeurIPS dataset submission track) and fine-tuned two generative models on the data. I do value the modifications the authors made to Wonder3D and the training ablation for Trellis, but it’s still a relatively small algorithmic contribution compared to the
level expected at this conference. That said, the work is still significant.



[1] Wang, Zehan and Zhang, Ziang and Pang, Tianyu and Du, Chao and Zhao,
Hengshuang and Zhao, Zhou, Orient Anything: Learning Robust Object Orientation
Estimation from Rendering 3D Models, ICML 2025




Minor typo:
Line 265: double period “Tesla A100 GPUs..”

---

> ### Author Rebuttal · Authors · 2025-07-31
>
> Thank you, Reviewer yWsi, for recognizing the significance of our work. Here we aim to address the concerns raised in the review.
>
> **W1. Limited theoretical/algorithmic contributions.**
>
> The main contributions of our work are that we are the first to identify the important but overlooked task, orientation-aligned 3D generation, via in-depth research on relevant works in the field of 3D generation. We demonstrate that the most effective solution is creating a new dataset and directly fine-tuning existing 3D generative models rather than commonly used post-processing methods, including PCA, Vision Language Model (VLM), and Orient Anything [1]. We also prove the importance of orientation-aligned 3D generation by firstly applying it to vital downstream applications.
>
> Besides, we also make several other innovations compared to relevant works. For the dataset curation, while the initial stage of our data curation pipeline is inspired by Orient Anything [1], we investigate and address the limitations of the VLM through manual inspection and correction, significantly improving dataset quality. For the orientation estimation application, we are the first to utilize an orientation-aligned 3D generative model for zero-shot model-free orientation estimation. This approach enhances the orientation awareness of SOTA 6D object pose estimation methods like Any6D [2], which use orientation-misaligned generative models for template creation. Our method also exhibits superior generalizability compared to SOTA orientation estimators like Orient Anything [1]. This is because we leverage the comprehensive 3D priors from powerful 3D generative models trained on larger orientation-misaligned datasets, in addition to the rich number of categories in our own dataset. For the object insertion application, we discover that 3D orientation alignment improves the efficiency of object orientation manipulation in simulation systems like the augmented reality application.
>
> [1] Wang, Zehan et al. "Orient Anything: Learning Robust Object Orientation Estimation from Rendering 3D Models." ICML 2025.
>
> [2] Lee, Taeyeop et al. "Any6D: Model-free 6D Pose Estimation of Novel Objects." CVPR 2025.
>
> **W2. Minor typo.**
>
> Thank you for pointing out our minor typo in Line 265. We will fix it in our final version.
>
> **Q1. More evaluation for their dataset construction pipeline.**
>
> Thank you for mentioning metrics utilized in other category-level pose canonicalization methods [3,4,5], including Instance-Level Consistency (IC), Category-Level Consistency (CC), Ground Truth Consistency (GC), and Ground Truth Equivariance Consistency (GEC). In fact, the evaluation illustrated in Figure 2 of our main paper is similar to calculating the IC metric used in [3,4,5] on datasets with ground truth.
>
> Specifically, our dataset curation pipeline comprises two stages: VLM recognition and subsequent manual correction. Consequently, the final curated dataset, Objaverse-OA, can be considered a dataset with manually annotated ground truth. In Figure 2, we evaluate the consistency between the VLM recognition results and our manually corrected Objaverse-OA results. Owing to its extensive number of categories, our Objaverse-OA is more suitable than other datasets with ground truth, like ShapeNet [6], for evaluating VLM performance on long-tailed categories.
>
> We now provide a detailed explanation of the minor difference between the IC metric and our evaluation metric. The IC metric calculates two-way Chamfer Distance between the canonicalized versions of the shapes (with superscript $^c$) before and after applying a set of random rotations $\mathbf{R}$ for shapes in the dataset $\mathcal{X}$:
>
> $$
> \mathrm{IC}:=\frac{1}{|\mathcal{X}||\mathbf{R}|}\sum_{X_i\in\mathcal{X}}\sum_{R_j\in\mathbf{R}}\mathrm{CD}[(R_jX_i)^c,X_i^c]
> $$
>
> Our metric also calculates the Chamfer Difference in the same way, but further applies a threshold of 0.01 to determine the correctness of recognition, as elaborated in the supplementary material section A.1:
>
> $$
> \mathrm{OurMetirc}:=\frac{1}{|\mathcal{X'}|}\sum_{X_i\in\mathcal{X'}} Thre(\mathrm{CD}[(R_iX_i)^c,X_i^c])
> $$
>
> $$
> Thre(CD) = \left\\{
> \begin{aligned}
> 1, \quad & \text{if } CD \leq 0.01 \\\\
> 0, \quad & \text{if } CD > 0.01
> \end{aligned}
> \right.
> $$
>
> As can be seen, the main difference between our metric and the IC metric is that we calculate recognition accuracy by thresholding the Chamfer Distance used in the IC metric. This is because the Chamfer Distance can be influenced by the mean shape of a category, potentially leading to an unfair comparison of VLM performance across different categories. Besides, instead of applying a set of user-defined random rotations $R$, our metric applies the original rotation $R_i$ in the Objaverse-LVIS dataset for each object $X_i$, which can lead to better evaluation of VLM's ability on the real misaligned dataset.
>
> Given the reviewer's suggestions, we looked into the other suggested metrics, CC, GC, and GEC. We believe the CC metric is not very relevant as it assesses the consistency of canonicalization across different instances within the same category, whereas our focus is on evaluating the VLM's cross-category canonicalization ability. The GC/GEC metric is also not essential since it measures canonicalization performance compared to manual labels, which is equivalent to our metric due to evaluation on our manually annotated dataset.
>
> [3] Agaram, Rohith et al. "Canonical Fields: Self-Supervised Learning of Pose-Canonicalized Neural Fields." CVPR 2023.
>
> [4] Sajnani, Rahul et al. "ConDor: Self-Supervised Canonicalization of 3D Pose for Partial Shapes." CVPR 2022.
>
> [5] Jin, Li et al. "One-shot 3D Object Canonicalization based on Geometric and Semantic Consistency." CVPR 2025.
>
> [6] Chang, Angel X. et al. “ShapeNet: An Information-Rich 3D Model Repository.” ArXiv 2015.
>
> **Q2: Concurrent work.**
>
> Thank you for mentioning the concurrent work that created the Canonical Objaverse Dataset [5]. We will cite this excellent work in our final version. The difference between our work and theirs lies in the alignment objective: our work aims to construct a cross-category aligned 3D dataset, whereas their focus is on creating an intra-category aligned 3D dataset.

---

> ### Comment · Reviewer_yWsi · 2025-08-03
>
> I thank the authors for the detailed response.
> While I'm still not completley concinced about the level of novelty, the rebuttal has largely addressed most of my concerns. I am increasing my rating from 4 to 5.

---

> > ### Author Response · Authors · 2025-08-04
> >
> > Dear Reviewer yWsi,
> >
> > We are glad to hear that our rebuttal addressed most of your concerns. Thanks again for your time and constructive suggestions. We will integrate these suggestions into our revision and further polish the presentation.

---

### Note · Authors · 2025-08-13

Dear AC and Reviewers,

We sincerely thank you for your hard work and valuable feedback on our submission.

First, we are grateful to the reviewers for recognizing the significance of our work and the proposed new task (Reviewers yWsi and y6ZQ), the comprehensiveness of our experiments (Reviewer MH83), and the merit of our downstream applications, which were described as compelling (Reviewer yWsi), practical (Reviewer y6ZQ), and interesting (Reviewer MH83).

During the rebuttal and discussion phases, we addressed the remaining concerns as follows:

- Reviewer yWsi: We clarified the novelty of our approach and the evaluation metrics used in our dataset construction pipeline.

- Reviewer y6ZQ: We clarified our definition and evaluation criteria for stick-like objects with ambiguous orientations, updated our evaluation metrics to avoid potential unfairness, and provided additional details on our method and experimental results.

- Reviewer MH83: We elaborated on the motivation behind our task, clarified the definition of our canonical space, and provided additional details on experimental results.

- Reviewer kr5a: We clarified our novelty, discussed concerns about realism, and conducted further analysis of VLM failure cases.

We greatly appreciate the reviewers' engagement and the AC's guidance during the rebuttal process. The constructive discussions have been invaluable in strengthening our work.

Thank you once again for your thoughtful evaluation and support.

Sincerely,

The Authors

---

### Decision · Program_Chairs · 2025-09-17

**Decision:**

Accept (poster)

**Comment:**

(a) This paper introduces Objaverse-OA, a large-scale dataset of 14,832 3D models annotated with orientation-aligned canonical poses across 1,008 categories. Using this dataset, the authors fine-tune two existing 3D generative models (Trellis and Wonder3D) to produce orientation-aligned outputs. These aligned models are then applied to two downstream tasks: 1) zero-shot object orientation estimation without requiring oriented CAD references, and 2) arrow-based object insertion into AR/VR environments. Extensive experimental results demonstrate the effectiveness of the proposed framework.

(b) The main contribution is a large and curated dataset of orientation-aligned 3D models that will likely be useful for future research. The paper is found to be easy to follow, with clear figures and useful supplementary materials. The reviewers appreciate the significance of the orientation alignment task and see the direct implications for robotics, AR/VR, and 3D content creation. The empirical validation demonstrates benefits for both orientation estimation and interactive object insertion.

(c) The main concerns are on the novelty. Since the main contribution is a dataset curation using an existing pipeline (Orient Anything), with limited algorithmic innovation beyond modest model modifications. Several reviewers view this as below the expected level of methodological novelty for NeurIPS. The task of estimating the orientation is not well-defined for all objects (e.g., spoons, sticks). There were concerns on the evaluation methodology (eg, there may be bias on the choice of viewpoints). There were also concerns on the data quality (eg, some generated objects appear toy-like or low-resolution). Reviewers found the downstream applications to have limitations (eg, insertion examples are more copy-paste rather than full integration).

(d) The dataset is substantial, well-annotated, and has clear practical value. The paper convincingly shows that orientation alignment improves downstream usability. The work will likely serve as a foundation for future research, particularly for robotics and AR/VR.
However, it is not considered for a spotlight nor an oral because the novelty is low (primarily data curation with light fine-tuning). Also, there are some fundamental issues with the task itself (orientation ground-truth is inherently ambiguous for some object classes). The technical contributions are not at the level of major algorithmic advance expected for a spotlight/oral.

(e) In the discussion and rebuttal three reviewers very overall satisfied with the clarifications, although the concerns on novelty and the ambiguities in the task largely remained. kr5a was the most negative reviewer (borderline reject). kr5a viewed the paper as primarily a dataset paper, but with limited realism and novelty.
In conclusion, the rebuttal addressed clarity and evaluation details but did not fully resolve concerns on novelty or orientation ambiguity. Nonetheless, three reviewers support acceptance, with consensus that the dataset and aligned models will have community value.